# BWS: Best Window Selection Based on Sample Scores for Data Pruning across Broad Ranges

## Abstract

Data subset selection aims to find a smaller yet informative subset of a large dataset that can approximate the full-dataset training, addressing challenges associated with training neural networks on large-scale datasets. However, existing methods tend to specialize in either high or low selection ratio regimes, lacking a universal approach that consistently achieves competitive performance across a broad range of selection ratios. We introduce a universal and efficient data subset selection method, Best Window Selection (BWS), by proposing a method to choose the best window subset from samples ordered based on their difficulty scores. This approach offers flexibility by allowing the choice of window intervals that span from easy to difficult samples. Furthermore, we provide an efficient mechanism for selecting the best window subset by evaluating its quality using kernel ridge regression. Our experimental results demonstrate the superior performance of BWS compared to other baselines across a broad range of selection ratios over datasets, including CIFAR-10/100 and ImageNet, and the scenarios involving training from random initialization or fine-tuning of pre-trained models.

## 1 Introduction

In many machine learning tasks, the effectiveness of deep neural networks often relies on large-scale datasets that include a vast number of samples, enabling them to achieve state-of-the-art performances. However, working with such large datasets presents several challenges, including the high computational costs, storage requirements, and potential concerns related to privacy (Schwartz et al., 2020; Strubell et al., 2019). A promising solution to mitigate these challenges is the concept of data subset selection. This approach involves the careful selection of a smaller, yet highly informative, subset extracted from the original large dataset. The goal is to find a subset with a specified selection ratio that approximates the performance of the entire dataset or incurs minimal performance loss.

Data subset selection has two primary approaches: the score-based selection and the optimization-based selection. In the score-based selection, a specific score is defined to quantify various aspects of each sample's influence (Koh & Liang, 2017), difficulty (Toneva et al., 2019; Paul et al., 2021), or consistency (Jiang et al., 2021) in training of neural networks. The primary goal is to identify the most valuable or influential samples within the dataset while pruning the remaining samples that have minimal impact on the model's generalization ability. On the other hand, optimization-based selection approaches find the optimal subset of a fixed size that can best approximate the full dataset training in terms of loss gradient or curvature by solving the associated optimization problem (Mirzasoleiman et al., 2020; Pooladzandi et al., 2022; Shin et al., 2023; Yang et al., 2023). The original optimization, which is NP-hard, is commonly approximated by submodular functions and a greedy algorithm is adopted to sequentially select the samples up to the size limit of the subset.

While the prior approaches successfully reduce dataset size in specific scenarios, there is not a single selection method that universally outperforms other baselins across broad selection ratios. To illustrate this, we conduct a benchmark comparison between two methods: Forgetting score (Toneva et al., 2019) representing the score-based selection approach, and LCMat (Shin et al., 2023) representing the optimization-based selection approach. We evaluate the test accuracy of models trained with different subset sizes of datasets, including CIFAR-10/100 (Krizhevsky, 2009) and ImageNet (Deng et al., 2009), ranging from 1% to 90%, as selected by these two methods (Table 1). Score-based methods, which prioritize samples of high influence or difficulty, tend to initially select rare

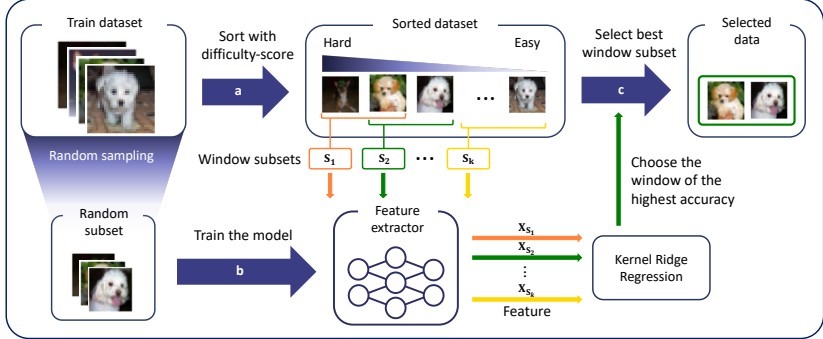

Figure 1: Overview of the proposed method, Best Window Selection (BWS). (a) Sort samples by a difficulty score (e.g., Forgetting (Toneva et al., 2019)) and generate window subsets of a fixed size while varying the starting point. (b) Train a feature extractor using random samples of the same size as the window subset only for a few epochs. (c) Extract the input features from each window subset and solve a kernel ridge regression. Evaluate the performance of the corresponding solution on the full training dataset to identify the best window subset achieving the highest accuracy.

yet influential samples while excluding typical or easy samples. These methods demonstrate competitive performance, nearly matching full-dataset training, when the selection ratio is sufficiently high (e.g., over 40% for CIFAR-10). However, they suffer significant performance degradation as the selection ratio decreases. In contrast, optimization-based methods tend to select representative samples that best approximate the full dataset training. Consequently, they achieve competitive performance even with very low selection ratios. However, their performance gains are limited as the selection ratio increases due to lack of diversity in sample selection. These findings show the variability in the criteria for an effective data subset, depending on the selection ratio, and highlight that previous methods may not be general enough to cover the entire spectrum of selection ratios.

Our key contribution in this paper is the development of a universal and efficient data selection method capable of maintaining competitive performance across a wide range of selection ratios. We introduce the Best Window Selection (BWS) method, illustrated in Fig. 1. The key idea involves ordering samples based on their difficulty-based sample scores and offering flexibility in choosing a window subset from the ordered samples, depending on the selection ratio and dataset. Specifically, we allow the starting point of each window subset to vary, enabling the selection of easy, moderate, or hard data subsets. We first demonstrate the existence of the best window that achieves the highest test accuracy for each subset size, and reveal that the optimal starting point for the best window varies depending on both the subset size and dataset. We then present a computationally-efficient method for selecting the best window subset without the need to evaluate models trained with each subset. We achieve this by solving a kernel ridge regression problem using samples from each window and evaluating the corresponding solution's performance on the full training dataset.

We evaluate our selection method, BWS on CIFAR-10/100 and ImageNet, and show that BWS consistently outperforms other baselines, including both score-based and optimization-based approaches, across a wide range of selection ratios ranging from 1% to 90%. For example, for CIFAR-10, BWS achieves a 30% improvement in test accuracy compared to Forgetting (Toneva et al., 2019) in the low selection ratios of 1-10%. It also demonstrates competitive performance in the high selection ratio regime, reaching up to 94% test accuracy with only a 40% data subset. Moreover, BWS consistently outperforms optimization-based techniques such as LCMat (Shin et al., 2023) and AdaCore Pooladzandi et al. (2022) across selection ratios from 5% to 75% for CIFAR-10.

## 2 RELATED WORKS

**Score-based selection**  Some initial works in score-based selection use a validation or test set to quantify the effect of each training sample. For instance, Data Shapley (Ghorbani & Zou, 2019; Kwon et al., 2021; Kwon & Zou, 2022) calculates the value of each data instance by measuring the average change in validation accuracy when that instance is excluded from the dataset. Influence

Table 1: Test accuracy across various selection ratios for the CIFAR-10/100 and ImageNet datasets, with subsets selected using random sampling, Forgetting score (Toneva et al., 2019), and LCMat (Shin et al., 2023). The best performance among the three is highlighted in **bold**.

| Selection ratio | | 1% | 5% | 10% | 20% | 30% | 40% | 50% | 75% | 90% |
|---|---|---|---|---|---|---|---|---|---|---|
| | Random | 49.59 | 77.35 | **84.14** | 89.15 | 91.10 | 92.41 | 93.29 | 94.60 | 95.01 |
| CIFAR-10 | Forgetting | 30.56 | 45.86 | 58.88 | 81.29 | 90.88 | **94.23** | **94.92** | **95.17** | **95.11** |
| | LCMat | **51.24** | **78.15** | 84.06 | **89.16** | **91.82** | 93.11 | 93.74 | 94.86 | 95.26 |
| | Random | 11.25 | 30.97 | 41.76 | 56.33 | **64.09** | **68.10** | 70.57 | 76.16 | 77.65 |
| CIFAR-100 | Forgetting | 11.71 | 23.19 | 34.32 | 48.83 | 59.11 | 66.18 | **71.67** | **77.43** | **78.33** |
| | LCMat | **16.16** | **35.21** | **46.80** | **57.25** | 63.28 | 67.82 | 71.74 | 76.66 | 78.01 |
| | Random | **6.14** | **33.17** | 45.87 | 59.19 | 65.94 | 68.23 | 70.14 | 73.74 | 74.83 |
| ImageNet | Forgetting | 4.78 | 28.18 | 45.84 | **60.75** | **67.48** | **70.26** | **72.73** | **74.63** | **75.53** |
| | LCMat | 6.01 | 32.26 | **46.08** | 59.02 | 65.28 | 68.50 | 70.30 | 74.13 | 74.81 |

Function (Koh & Liang, 2017; Pruthi et al., 2020) approximates how a model's prediction changes as individual training examples are visited. In the absence of a validation set, score-based selection quantifies the difficulty or consistency of samples during neural network training. Forgetting (Toneva et al., 2019) and EL2N (Paul et al., 2021) introduce a difficulty score to measure a data point's learning difficulty. Memorization (Feldman & Zhang, 2020) and c-score (Jiang et al., 2021) aim to predict the accuracy of a sample when the full dataset is utilized, except for that sample. CG-score (Ki et al., 2023) evaluates data instances without model training by calculating the analytical gap in generalization errors when an instance is held out. These score-based methods prioritize difficult or influential samples for data subset selection. While they effectively select a subset approximating the full-dataset performance, their performance degrades significantly as the selection ratio decreases, as achieving high performance solely with difficult samples becomes challenging.

**Optimization-based selection** Optimization-based selection involves formulating an optimization problem to select a coreset of a given size that can effectively approximate the diverse characteristics of the full dataset. These methods include coreset selection to approximate the training distribution by herding (Chen et al., 2010) or k-center greedy algorithms (Sener & Savarese, 2018). Recent approaches have also sought subsets of samples that can approximate loss gradients or curvature by CRAIG (Mirzasoleiman et al., 2020), CREST (Yang et al., 2023), and AdaCore (Pooladzandi et al., 2022). While these methods have proven effective, they are computationally demanding and necessitate full-dataset sampling at each epoch. LCMat (Shin et al., 2023) addresses this computational challenge by aligning both gradients and Hessians without requiring periodic full-dataset sampling. However, these methods often struggle to choose diverse samples, and their performance does not match that of score-based approaches, in the intermediate to high selection ratio regimes.

In contrast to previous approaches, we develop a universal selection method capable of consistently identifying a high-performance subset across a wide range of selection ratios. While recent methods like Moderate-DS (Xia et al., 2023) and CCS (Zheng et al., 2023) have also aimed for universality across various selection ratios, our method outperforms these approaches, over a broad range of selection ratios, as demonstrated in Section 5. Moderate-DS selects samples closest to the median of the features of each class, while CCS prunes a $\beta\%$ of hard examples, with $\beta$ being a hyperparameter, and then selects samples with a uniform difficulty score distribution. Importantly, our method does not require hyperparameter tuning, such as setting $\beta$ in CCS. This is because we propose a method to assess the quality of window subsets and efficiently find the best one using kernel ridge regression.

## 3 MOTIVATION

We conduct an evaluation of existing data selection methods across a wide range of selection ratios. Specifically, we benchmark two representative methods: Forgetting score (Toneva et al., 2019), representing difficulty score-based selection, and LCMat (Shin et al., 2023), representing optimization-based selection. We assess the test accuracy of models trained on subsets of CIFAR-10/100 and ImageNet, with selection ratios ranging from 1% to 90%, as summarized in Table 1. For the Forgetting score approach, we sort the samples in descending order based on their scores, defined as the

number of times during training the decision of that sample switches from a correct one to incorrect one, and select the top-ranking (most difficult) samples. In contrast, for LCMat, we employ an optimization to identify a subset that best approximates the loss curvature of the full dataset. We employ ResNet18 (He et al., 2016) for CIFAR-10 and ResNet50 for CIFAR-100 and ImageNet.

We can observe that the most effective strategy varies depending on the selection ratios, and there is no single method that consistently outperforms others across the entire range of selection ratios. Specifically, for CIFAR-10 with low subset ratios (1-30%), the optimization-based selection (LCMat) performs better than the difficulty score-based selection (Forgetting). In this regime, the 'Forgetting' even underperforms random selection. However, as the subset ratio increases beyond 40%, the 'Forgetting' outperforms both the LCMat and random selection. Similar trends are observed for CIFAR-100 and ImageNet. Interestingly, for CIFAR-100, there is an intermediate regime where neither the 'Forgetting' nor LCMat outperform the simplest random sampling.

These findings emphasize that the desired properties of data subsets change depending on the selection ratios. In cases of low selection ratios (sample-deficient regime), it is more beneficial to identify a representative subset that closely resembles the full dataset in terms of average loss gradients or curvature during training. However, as the selection ratio increases (sample-sufficient regime), preserving the high-scoring, rare or difficult-to-learn samples becomes more critical, as these samples are known to enhance the generalization capability of neural networks and cannot be fully captured by a representative subset that reflects only the average behavior of the full dataset (Ki et al., 2023).

## 3.1 THEORETICAL ANALYSIS

To validate this experimental finding, we further provide a theoretical analysis of optimal subset selection, which reveals similar change of trends in the desirable subsets depending on the selection ratios. We consider a binary classification problem and the problem setup is summarized below:

- Data samples $\mathbf{x}_1, \mathbf{x}_2, \ldots \mathbf{x}_n \in \mathbb{R}^d$ are generated from a multivariate normal distribution, $\mathcal{D} = \frac{1}{\sqrt{d}}\mathcal{N}(0, \mathbf{I}_d)$. The label $y_i$ of sample $\mathbf{x}_i$ is determined by the sign of its first element. Specifically, if $(\mathbf{x}_i)_1 > 0$ then $y_i = 1$, and if $(\mathbf{x}_i)_1 < 0$, then $y_i = -1$. We define the score of each sample as $1/|(\mathbf{x}_i)_1|$. Samples closer to the decision boundary $(\mathbf{x})_1 = 0$ have higher scores, while those farther from the boundary have lower scores.

- We select a label-balanced subset of size $m$, denoted by $(\mathbf{X_S}, \mathbf{y_S}) \in \mathbb{R}^{d \times m} \times \{-1, 1\}^m$, and use it to solve the linear regression problem to find $\mathbf{w_S} = \arg\min_{\mathbf{w} \in \mathbb{R}^d} ||\mathbf{y_S} - \mathbf{X_S}^\top \mathbf{w}||_2^2$. For a new sample $\mathbf{x}'$, our decision will be $+1$ if $\mathbf{w_S}^\top \mathbf{x}' > 0$ and $-1$ otherwise. Therefore, we consider $\mathbf{w_S}$ to be a better solution when the value of its first element, $(\mathbf{w_S})_1$, is larger.

For the above setup, we analyze the solution $\mathbf{w_S}$ depending on the subset size $|\mathbf{S}|$.

**Theorem 1** (Informal). *If the subset size is as small as $|\mathbf{S}| = m \ll \sqrt{d/\ln d}$, then the first coordinate of $\mathbf{w_S}$ is approximated as $(\mathbf{w_S})_1 \approx \sum_{i=1}^{m} |(\mathbf{x}_i)_1|$. On the other hand, if $|\mathbf{S}| = m \gg d^2 \ln d$, it can be approximated as $(\mathbf{w_S})_1 \approx (\sum_{i=1}^{m} |(\mathbf{x}_i)_1|)/(\sum_{i=1}^{m} |(\mathbf{x}_i)_1|^2)$.*

A more formal statement and the proof of Thm. 1 is available in Appendix A.2. From Thm.1, it is evident that the characteristics of the desirable data subset $\mathbf{X_S}$ vary depending on the subset size regime. In the sample-deficient regime ($m \ll \sqrt{d/\ln d}$), it is more advantageous to include samples that are farther from the decision boundary (easy samples) in $\mathbf{X_S}$ to train a better classifier, resulting in a higher value of $(\mathbf{w_S})_1$. Conversely, in the sample-sufficient regime ($m \gg d^2 \ln d$), it is more beneficial to include samples closer to the decision boundary (difficult samples) to increase $(\mathbf{w_S})_1$. We conjecture that the relatively wide gap between two distinct regimes ( $[\sqrt{d/\ln d}, d^2 \ln d]$ ) may be attributed to the loose analysis. We anticipate that a more precise boundary will occur at $m = \Theta(d)$, where $m \ll d$ ($m \gg d$) corresponds to the sample-deficient (sufficient) regime. We provide empirical results that support this theoretical analysis and our conjecture in Appendix A.3.

Having identified the distinct properties of desirable data subsets depending on the subset size, the remaining question is how to design a universal data selection method capable of performing well across a wide range of sample selection ratios. To address this question, we explore the feasibility of a simple yet efficient approach: window selection with varying starting points, wherein the data samples are ordered based on their difficulty scores.

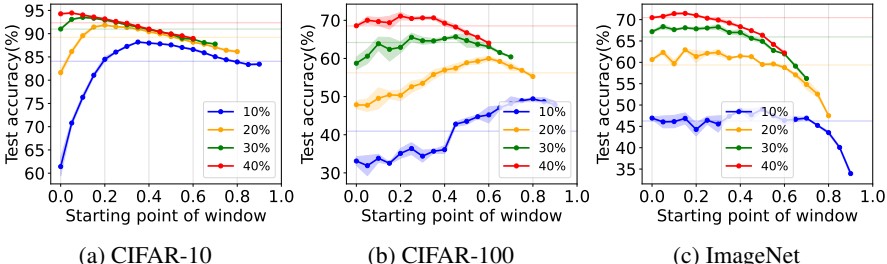

(a) CIFAR-10       (b) CIFAR-100       (c) ImageNet

Figure 2: Sliding window experiments to measure the test accuracy of the models trained by window subsets while changing the starting point of the windows. Samples are sorted in descending order by their difficulty scores. The horizontal lines are results from random selection. For each subset ratio, there exists the best window, and its starting point shifts toward left as the subset ratio increases.

# 4 METHODOLOGY

To develop a universal method capable of performing effectively across a wide range of sample selection ratios, we consider a window selection method of varying width, depending on the selection ratio, applied to the samples ordered according to their difficulty scores. This approach has two merits: 1) flexibility and 2) computational-efficiency. By sorting the samples in descending order based on their difficulty scores and selecting a starting point, such as $s\%$ for a given window size of $w\%$, we can choose continuous intervals of samples within $[s, s + w]\%$. This flexibility allows us to opt for easy, moderate, or hard data subsets depending on the choice of the starting point. Moreover, the search space of window selection method is confined to the number of possible starting points for the windows, making the window selection method computationally much more efficient compared to a general subset selection where the search space scales as $\binom{n}{m} \approx \exp(cn)$ for some constant $c > 0$ when the subset size $m$ is a constant fraction of $n$.

We first explore the performance of the window selection approach while varying the starting point and illustrate the existence of the best window subset. We sort the samples from CIFAR-10/100 and ImageNet in descending order based on their Forgetting scores (Toneva et al., 2019), and select windows of different sizes, ranging from $10\%$ to $40\%$, by adjusting the starting point from 0 to $(100 - w)\%$ with a step size of $5\%$. We then train ResNet18 for CIFAR-10 and ResNet50 for CIFAR-100/ImageNet using the windows subsets and plot the resulting test accuracies in Fig. 2.

We can observe that, for each subset ratio, there exists an optimal starting point, and this optimal point shifts towards lower values (indicating more difficult samples) as the window subset size increases. Specifically, for CIFAR-10, the optimal window subset of size $10\%$ falls within the interval $[35, 45]\%$, while for a window size of $40\%$, it falls within $[5, 45]\%$. Similar trends are observed for CIFAR-100 and ImageNet, albeit with distinct optimal starting points depending on the dataset. For CIFAR-100, with a window size of $10\%$, the best window subset comprises samples from $[80, 90]\%$, primarily consisting of easy samples. It is important to note that the $10\%$ subset for CIFAR-100 includes only 50 samples per class, whereas for CIFAR-10, it includes 500 samples per class. Consequently, the optimal $10\%$ window for CIFAR-100 ($[80, 90]\%$) tends to include more easy and representative samples capable of capturing the major characteristics of each class.

The observation that the optimal starting point of the window subset varies based on both the subset size and the dataset introduces a new challenge in window selection: How can we efficiently identify the best window subset without having to evaluate models trained on each subset? We address this crucial question by introducing a proxy task to estimate the quality of window subsets.

## 4.1 BWS: BEST WINDOW SELECTION

Our goal is to develop a computationally-efficient method capable of assessing and identifying the best window subset without requiring the training of a model on every potential subset. To achieve this goal, we propose to solve a kernel ridge regression problem by using each window subset and evaluate the performance of the corresponding solution on the full training datasets. Algorithm 1 outlines the specific steps involved in this process.

---

**Algorithm 1** BWS: Best Window Selection Method

---

**Input** Dataset $\{(\mathbf{x}_i, y_i)\}_{i=1}^n$ sorted by difficulty-based sample scores $\{s_i\}_{i=1}^n$ in descending order, subset size $m$, and step size $t$.

    Train a feature extractor $f(\cdot)$ by $m$ randomly chosen samples from the dataset.
    Extract the features of the samples by using $f(\cdot)$ and denote them by $\mathbf{f}_i = [f(\mathbf{x}_i), 1]$.
    **for** $k \in \{0, t, 2t, 3t \ldots, \lfloor (n-m)/t \rfloor t\}$ **do**
        Define a window subset $\mathbf{S} = \{(\mathbf{f}_i, y_i)\}_{i=k}^{k+m-1}$.
        **for** $c \in \{1, 2, \ldots C\}$ **do**
            For the samples in $\mathbf{S}$ with label $c$, set the label equal to 1. For others, set the label to 0.
            Solve the linear regression problem Eq.1 with the window subset $\mathbf{S}$. Let $\mathbf{w_S}(c)$ be the solution.
        **end for**
        Obtain $\mathbf{w_S} \in \mathbb{R}^{C \times (d+1)}$ by defining $\mathbf{w_S} := [\mathbf{w_S}(1), \ldots \mathbf{w_S}(C)]$.
        Calculate the accuracy of $\mathbf{w_S}$ by $\frac{1}{n} \sum_{i=1}^n \mathbb{1}(\arg\max_c (\mathbf{w_S}^\top \mathbf{f}_i)_c = y_i)$.
    **end for**

**Output** Window subset $\mathbf{S}$ for which the accuracy of $\mathbf{w_S}$ is maximized.

---

Let $\mathbf{f}_i := [f(\mathbf{x}_i), 1] \in \mathbb{R}^{d+1}$ be the feature vector of $\mathbf{x}_i$ obtained by a feature extractor $f(\cdot)$. The details of the feature extractor is available in the end of this section. For each window subset $\mathbf{S} = \{(\mathbf{f}_i, y_i)\}_{i=1}^m$ composed of $m$ samples, define $\mathbf{X_S} := [\mathbf{f}_1, \ldots, \mathbf{f}_m]$ and $\mathbf{y_S} := [y_1, \ldots, y_m]$. Then, we denote the problem of kernel ridge regression, and the corresponding solution, using the subset $\mathbf{S}$ by

$$\mathbf{w_S} := \arg\min_{\mathbf{w}} \sum_{(\mathbf{f}_i, y_i) \in \mathbf{S}} (y_i - \mathbf{w}^\top \mathbf{f}_i)^2 + \lambda \|\mathbf{w}\|^2 = \arg\min_{\mathbf{w}} \|\mathbf{y_S} - \mathbf{X_S}^\top \mathbf{w}\|_2^2 + \lambda \|\mathbf{w}\|^2, \quad (1)$$

$$\mathbf{w_S} = (\lambda \mathbf{I}_{d+1} + \mathbf{X_S} \mathbf{X_S}^\top)^{-1} \mathbf{X_S} \mathbf{y_S} = \mathbf{X_S} (\lambda \mathbf{I}_m + \mathbf{X_S}^\top \mathbf{X_S})^{-1} \mathbf{y_S}. \quad (2)$$

We set $\lambda = 1$ to prevent singularity in matrix inversion. The matrix inversion in Eq. 2 can be performed efficiently in a lower dimension between $d+1$ and $m$.

Our algorithm finds the best window subset by evaluating the performance of $\mathbf{w_S}$, corresponding to each window subset $\mathbf{S}$, on classifying the training samples $\{(\mathbf{x}_i, y_i)\}_{i=1}^n$ as described in Alg. 1. To apply $\mathbf{w_S}$ for $C$-class classification problem, we find $\mathbf{w_S}(c) \in \mathbb{R}^{d+1}$ for each class $c \in \{1, \ldots, C\}$, classifying whether a sample belongs to class $c$ or not, and simply place the vectors in columns of $\mathbf{w_S} \in \mathbb{R}^{C \times (d+1)}$. Then, we evaluate the performance of $\mathbf{w_S}$ by calculating the classification accuracy $\frac{1}{n} \sum_{i=1}^n \mathbb{1}(\arg\max_c (\mathbf{w_S}^\top \mathbf{f}_i)_c = y_i)$ of $\mathbf{w_S}$ on the full training dataset.

In Table 2, we compare the performances of window subsets of CIFAR-10 with different starting points, in terms of their 1) test accuracy, measured on models actually trained with the window subsets and 2) accuracy of kernel ridge regression on the full training dataset, serving as a proxy for evaluating the subset's quality. The results show a strong alignment between the best-performing windows, as indicated by both performance measures, for each subset ratio. This observation demonstrates the effectiveness of our algorithm, which can efficiently replace the need to train models on each window subset and evaluate them on test dataset. Results for CIFAR-100 dataset and ImageNet dataset are also provided in Appendix H.

**Feature extractor** When $|\mathbf{S}| = m$, we randomly choose $m$ samples from the full dataset, and use these samples to train a neural network only for a few epochs to generate a feature extractor $f(\cdot)$. For CIFAR-10 dataset, we train ResNet18 for 20 epochs, and for CIFAR-100 and ImageNet, we train ResNet50 for 20 epochs. The rationale behind training a feature extractor with random samples that match the window subset's size is to simulate the scenario where the model is trained using the restricted window subset of that size, enabling effective quality evaluation for window subsets.

**Computational complexity** The computational complexity of Algorithm 1 consists of training a feature extractor and solving the regression problem for $(\lfloor (n-m)/t \rfloor t)$-subsets. Feature extractor training is relatively efficient since it involves only a few epochs. Solving the regression requires matrix inversion, which takes $O(d^3)$ steps, with $d = 512$ for ResNet18 and 2048 for ResNet50. Detailed computational times are provided in Appendix C.3.

Table 2: Comparison of window subsets of CIFAR-10 in terms of their 1) test accuracy, measured on models trained with the window subsets (top rows) and 2) accuracy of kernel ridge regression on the training dataset (bottom rows). The best performing windows align well between the two measures.

| Ratio | Starting point | 0% | 5% | 10% | 15% | 20% | 25% | 30% | 35% | 40% | 50% | 70% | 90% |
|---|---|---|---|---|---|---|---|---|---|---|---|---|---|
| 10% | Test Acc | 60.62 | 70.52 | 76.03 | 80.81 | 84.17 | 85.74 | 87.06 | **88.05** | 87.82 | 87.28 | 84.74 | 82.95 |
| | Regression Acc | 56.84 | 61.98 | 64.30 | 65.61 | 66.73 | 67.40 | 67.96 | **68.38** | 68.35 | 68.07 | 67.70 | 67.40 |
| 20% | Test Acc | 81.35 | 85.84 | 89.38 | 91.34 | **91.69** | 91.29 | 91.15 | 90.85 | 90.18 | 89.30 | 86.83 | - |
| | Regression Acc | 76.43 | 77.93 | 78.90 | 79.53 | 79.93 | **79.96** | 79.95 | 79.89 | 79.71 | 79.44 | 78.87 | - |
| 30% | Test Acc | 90.84 | 92.82 | **93.35** | 93.17 | 92.82 | 92.31 | 91.63 | 91.28 | 90.72 | 89.66 | 87.51 | - |
| | Regression Acc | 83.75 | 84.44 | 84.71 | **84.74** | 84.70 | 84.58 | 84.48 | 84.33 | 84.18 | 83.97 | 83.64 | - |
| 40% | Test Acc | 94.11 | **94.38** | 93.93 | 93.46 | 93.03 | 92.50 | 92.09 | 91.38 | 90.92 | 89.86 | - | - |
| | Regression Acc | 87.91 | **87.93** | 87.88 | 87.76 | 87.63 | 87.51 | 87.34 | 87.20 | 87.04 | 86.86 | - | - |

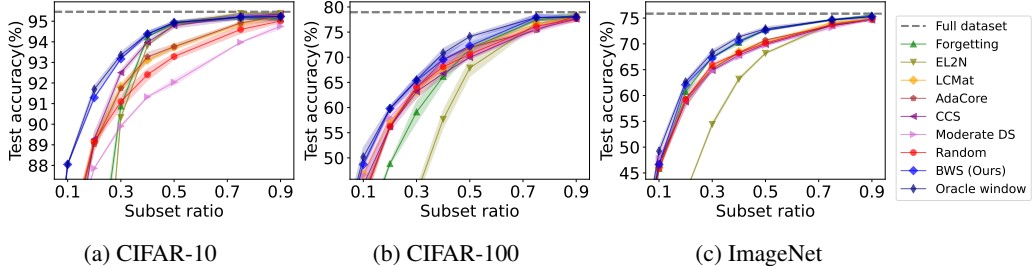

(a) CIFAR-10        (b) CIFAR-100        (c) ImageNet

Figure 3: Test accuracy of the models trained with data subsets of varying ratios, selected by different methods. Our method (BWS) outperforms other baselines across a wide range of selection ratios and achieves the accuracy as high as the Oracle window. Full results are reported in Table 13–15.

# 5 EXPERIMENTS

To demonstrate the effectiveness of our method, we conduct data pruning experiments similar to those in (Shin et al., 2023; Zheng et al., 2023). We select a subset of the dataset using each selection method while pruning the rest of the samples, and evaluate the performance of the model trained with each subset. We perform these experiments on CIFAR-10/100 and ImageNet, using ResNet18 for CIFAR-10 and ResNet50 for CIFAR-100/ImageNet. As baselines, we include 1) two difficulty score-based selection methods: Forgetting (Toneva et al., 2019) and EL2N (Paul et al., 2021), 2) two optimization-based selection methods: AdaCore (Pooladzandi et al., 2022) and LCMat (Shin et al., 2023), and 3) two universal selection methods: Moderate DS score (Xia et al., 2023) and CCS (Zheng et al., 2023). More details about the baselines and experiments are available in Appendix C. The full experimental results of this section are provided in Appendix H.

## 5.1 EXPERIMENTAL RESULTS

**Data pruning experiments** In Fig. 3, we present the test accuracies of models trained with data subsets of varying ratios, selected by different methods. The reported values are mean, and the shaded regions represent the standard deviation across three (two) independent runs for CIFAR-10/100 (ImageNet). The gray dotted lines represent the results with the full dataset, while the red curve represents the results of random selection. The Oracle window curve represents the results obtained using the window subset of the highest test accuracy found by the sliding window experiment as in Fig.2, and BWS represents the results obtained using Alg.1. From the results, we observe that our method, BWS, consistently outperforms all other baselines across almost all selection ratios, and achieves the performance near that of the Oracle window. In the case of CIFAR-10/100, the difficulty score-based methods, Forgetting and EL2N, perform well in high ratio regimes but experience significant performance degradation as the selection ratio decreases, while Forgetting still maintains competitive performance in ImageNet. The optimization-based methods, LCMat and AdaCore, achieve better performance than the difficulty score-based methods for low selection ratios but underperform in high selection ratios. The two previous universal selection methods, Moderate DS and CCS, also underperform compared to ours across almost all selection ratios.

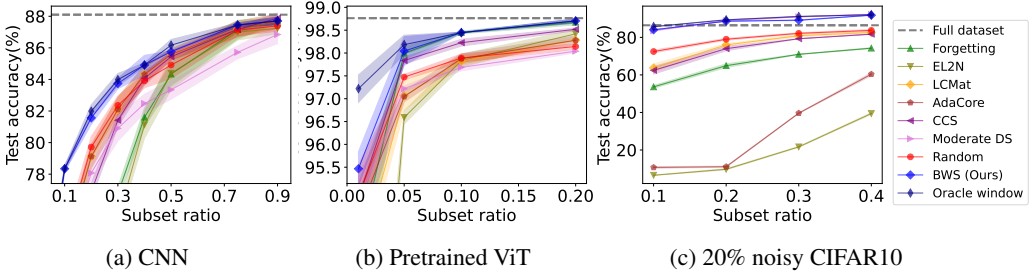

(a) CNN        (b) Pretrained ViT        (c) 20% noisy CIFAR10

Figure 4: (a, b) Cross-architecture experiments with CNN (left), and pre-trained ViT (middle), where samples scores are calculated using ResNet18 model. (c) Data pruning experiments with CIFAR-10, including 20% label-noise. Full results are reported in Table 16–19.

**Cross-architecture robustness** To test the robustness of our method across changes in neural network architectures, we conduct data pruning experiments on CIFAR-10 while using different architectures during sample scoring and training. The window subsets are constructed using samples ordered by their Forgetting scores, calculated on ResNet18 architecture. Then, the best window selection (Alg. 1) and the model training are conducted using a simpler CNN architecture or a larger network, Vision Transformer (ViT) (Dosovitskiy et al., 2021), pre-trained on the ImageNet dataset. The results on the CNN architecture are presented in Fig. 4a, while those on ViT are shown in Fig. 4b. In both cases, our method (BWS) consistently achieve competitive performances across all selection ratios, demonstrating its robustness to changes in neural network architectures during data subset selection. For the ViT results, using only about 5% of the CIFAR-10 dataset selected by BWS achieves a test accuracy of 98.04%, comparable to the test accuracy of 98.60% achievable with the full dataset. This result also demonstrates the effectiveness of our method in selecting samples for fine-tuning of a pre-trained model. Additional results using another model, EfficientNet-B0 (Tan & Le, 2019), are reported in Appendix G.1.

**Robustness to label noise** We test the robustness of BWS in the presence of label noise in the training dataset. We corrupt randomly chosen 20% samples of CIFAR-10 by random label noise. It has been previously reported that the difficulty score-based selection methods are susceptible to label noise since such methods tend to assign high scores to label-noise samples (Toneva et al., 2019; Paul et al., 2021). Thus, these methods often ends up prioritizing the label-noise samples in the selection process, leading to suboptimal results. On the other hand, our algorithm offers flexibility in choosing window subsets with varying levels of difficulty by changing the starting point, and adopts an approach to select the best window by solving a proxy task using the kernel ridge regression. To further enhance the robustness of our method, we can modify Alg. 1 to evaluate the solution of kernel ridge regression using only the low-scoring 50% samples from the training dataset, which will rarely include label-noise samples, instead of the full dataset. In Fig. 4c, we compare the performance of this modified version of BWS with other baselines. While difficulty score-based selection and optimization-based selection methods suffer from performance degradation due to label noise, our method, along with another label noise-robust method, Moderate DS, achieves performance even higher than what is achievable with the full training dataset, which includes the 20% label noise. This demonstrates the effectiveness and robustness of our approach in handling label noise.

## 5.2 Ablation study

Our BWS algorithm operates by sorting the training data samples based on their difficulty scores, creating window subsets, and then selecting the best window subset by a proxy task. To assess the relative importance of each component, we conduct several ablation studies in this section.

**Different types of windows** Our method considers a window type consisting of samples from a continuous interval of difficulty scores while varying the starting point. We explore four different variations of window types: 1) Hard-only, which involves the subset selection composed of the highest scoring (most difficult) samples, 2) Easy-only, which involves the subset selection composed of the lowest scoring (easiest) samples, 3) Hard-easy, which balances the selection by choosing an

Table 3: Test accuracy of the models trained by different types of window subsets of CIFAR-10.

| Selection ratio | Hard-only | Easy-only | Hard-easy | 25-75% | BWS (ours) |
|---|---|---|---|---|---|
| 10% | 60.62 | 82.95 | 73.32 | 87.17 | **88.05** |
| 20% | 81.35 | 85.99 | 81.42 | 89.96 | **91.29** |
| 30% | 90.84 | 87.51 | 87.44 | 91.11 | **93.17** |
| 40% | 94.11 | 88.80 | 90.82 | 91.77 | **94.38** |

Table 4: Test accuracy of the models trained by window subsets of CIFAR-10 selected by different strategies in choosing the best window subset. Our method consistently achieves the better performance, and the best window subsets selected by ours aligns better with those of oracle windows.

| Selection methods | Selection ratio | 1% | 5% | 10% | 20% | 30% | 40% | 50% | 75% | 90% |
|---|---|---|---|---|---|---|---|---|---|---|
| Gradient $\ell_2$-norm difference | Test accuracy | 54.78 | 81.79 | 87.82 | 90.85 | 91.63 | 92.50 | 93.07 | 94.36 | 94.92 |
| | Window index | 50% | 45% | 40% | 35% | 30% | 25% | 20% | 10% | 5% |
| Gradient cosine similarity | Test accuracy | 43.39 | 71.99 | 84.17 | 91.34 | 93.35 | 94.38 | 94.93 | 95.20 | 95.22 |
| | Window index | 30% | 25% | 20% | 15% | 10% | 5% | 0% | 0% | 0% |
| Regression (ours) | Test accuracy | 63.14 | 81.31 | 88.05 | 91.29 | 93.17 | 94.38 | 94.93 | 95.20 | 95.22 |
| | Window index | 90% | 70% | 35% | 25% | 15% | 5% | 0% | 0% | 0% |
| Oracle window | Test accuracy | 65.73 | 83.03 | 88.05 | 91.69 | 93.35 | 94.38 | 94.93 | 95.20 | 95.22 |
| | Window index | 80% | 50% | 35% | 20% | 10% | 5% | 0% | 0% | 0% |

equal number of highest-scoring and lowest-scoring samples, 4) 25-75%, which involves random selection from a window subset spanning 25 to 75% score-ranked samples. Table 3 summarizes the resulting test accuracies when the subsets are selected from CIFAR-10 by using each window type, with selection ratios ranging from 10 to 40%. Our method consistently achieves better performance compared to all the variations. Simply mixing the most difficult and easiest samples (Hard-easy) or randomly sampling from a moderate regime (25-75%) does not yield as strong results as our method.

**Different window selection methods**    We also evaluate the effectiveness of our window selection strategy in Alg. 1 based on kernel ridge regression by comparing it with two different variants in the best window selection: 1) Gradient $\ell_2$-norm difference, which aims to find a window subset that minimizes the $\ell_2$-norm of the difference between the average gradients of the full dataset and the window subset, and 2) Gradient cosine similarity, which aims to find a window subset that maximizes the cosine similarity between the average gradients of the full dataset and the window subset. These methods are inspired by gradient-matching strategies used in optimization-based coreset selection (Mirzasoleiman et al., 2020; Yang et al., 2023). Table 4 presents the test accuracies achieved by models trained on window subsets selected by each method, along with the corresponding starting point of the best window chosen by each method. The last row shows the result with the oracle window. Our method consistently achieves better test accuracy compared to the two variants, and the window selected by our method aligns better with the oracle selection. This result demonstrates that the best subset cannot be effectively chosen by simply matching the average gradients of the full training dataset; it requires a proxy task such as kernel ridge regression to evaluate the quality of window subsets for classification tasks.

We also perform an additional ablation study to show the robustness of our method across various difficulty scores used for ordering the samples in Appendix G.2.

## 6    CONCLUSION

We introduced the Best Window Selection (BWS), a universal and efficient data subset selection method capable of achieving competitive performance across a wide range of selection ratios. This represents a notable improvement over previous data subset selection methods, which typically excel within a restricted range of selection ratios. Our experimental results demonstrate that BWS effectively identifies the best window subset from samples ordered by difficulty-based score, by leveraging a simple proxy task based on kernel ridge regression.

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

# A    PROOF OF THEORETICAL ANALYSIS

## A.1    LINEAR RIDGE REGRESSION

The solution of the linear ridge regression problem is derived as follows.

$$L(\mathbf{w}) = ||\mathbf{y} - \mathbf{X}^\top \mathbf{w}||_2^2 + \lambda ||\mathbf{w}||_2^2$$

$$\frac{\partial L}{\partial \mathbf{w}} = 2\mathbf{X}\mathbf{X}^\top \mathbf{w} - 2\mathbf{X}\mathbf{y} + 2\lambda \mathbf{w} = 0 \quad \Rightarrow \quad \mathbf{w} = (\lambda \mathbf{I} + \mathbf{X}\mathbf{X}^\top)^{-1}\mathbf{X}\mathbf{y}$$

$$\therefore \mathbf{w_S} = (\lambda \mathbf{I} + \mathbf{X_S}\mathbf{X_S}^\top)^{-1}\mathbf{X_S}\mathbf{y_S} = \mathbf{X_S}(\lambda \mathbf{I} + \mathbf{X_S}^\top \mathbf{X_S})^{-1}\mathbf{y_S}$$

## A.2    PROOF OF THEOREM 1

In this section, we provide the detailed proof of Theorem 1 in Sec. 3.1. Note that $n = poly(d)$ data inputs $\mathbf{x}_1, \mathbf{x}_2, \ldots \mathbf{x}_n$ are sampled from normalized multivariate normal distribution, $\mathcal{D} = \frac{1}{\sqrt{d}}\mathcal{N}(0, \mathbf{I}_d) = \frac{1}{\sqrt{d}}(\mathcal{N}_1, \mathcal{N}_2 \ldots \mathcal{N}_d)$ where $\{\mathcal{N}_k\}$ are $i.i.d.$ normal distributions. Remind that the label $y_i$ of sample $\mathbf{x}_i$ is determined by the sign of its first element, i.e., if $(\mathbf{x}_i)_1 > 0$ then $y_i = 1$, and if $(\mathbf{x}_i)_1 < 0$, then $y_i = -1$. We select a subset of size $m$, denoted by $(\mathbf{X_S}, \mathbf{y_S}) \in \mathbb{R}^{d \times m} \times \{-1, 1\}^m$.

We first provide a high-level proof idea of Theorem 1. Note that the optimal $\mathbf{w_S} = \arg\min_{\mathbf{w} \in \mathbb{R}^d} ||\mathbf{y_S} - \mathbf{X_S}^\top \mathbf{w}||_2^2$ can be written as $\mathbf{w_S} = \mathbf{X_S}(\mathbf{X_S}^\top \mathbf{X_S})^{-1}\mathbf{y_S}$ when $m \leq d$, and $\mathbf{w_S} = (\mathbf{X_S}\mathbf{X_S}^\top)^{-1}\mathbf{X_S}\mathbf{y_S}$ when $m \geq d$. Let us first consider the case of $m \leq d$. Due to the properties of high dimensional multivariate normals, we have $||\mathbf{x}_i|| \in [1 \pm \sqrt{7 \ln n / 2d}]$ for all $i \in [n]$ and $|\mathbf{x}_i^\top \mathbf{x}_j| \leq \sqrt{7 \ln n / 2d}$ for all $i \neq j \in [n]$ with high probability. Thus, $||\mathbf{X_S}^\top \mathbf{X_S} - \mathbf{I}_m||_F \leq \sqrt{\frac{m^2(7 \ln n)}{2d}}$ where $\mathbf{I}_m$ is the identity matrix of size $m$. When $m \ll \sqrt{d / \ln d}$, we have $(\mathbf{X_S}^\top \mathbf{X_S}) \approx (\mathbf{X_S}^\top \mathbf{X_S})^{-1} \approx \mathbf{I}_m$, and thus $\mathbf{w_S} = \mathbf{X_S}(\mathbf{X_S}^\top \mathbf{X_S})^{-1}\mathbf{y_S} \approx \mathbf{X_S}\mathbf{y_S}$, which implies that $(\mathbf{w_S})_1 \approx \sum_{i=1}^m |(\mathbf{x}_i)_1|$. Let us next consider the case of $m \geq d$. Note that the diagonal terms of $\mathbf{X_S}\mathbf{X_S}^\top$ are $\sum_{i=1}^m |(\mathbf{x}_i)_k|^2 = \Theta(m/d)$ for $k \in [d]$ and the off-diagonal terms are $\sum_{i=1}^m (\mathbf{x}_i)_k(\mathbf{x}_i)_l = O(\sqrt{m \ln d}/d)$ for $k \neq l \in [d]$ with high probability. The eigenvalues of $\mathbf{X_S}\mathbf{X_S}^\top$ can be shifted from its diagonal entries $(\sum_{i=1}^m |(\mathbf{x}_i)_1|^2, \ldots, \sum_{i=1}^m |(\mathbf{x}_i)_d|^2)$ by at most $\frac{\sqrt{m \ln d}}{d}d = \sqrt{m \ln d}$ by the effect of its off-diagonal entries. Thus, when $m/d \gg \sqrt{m \ln d}$, i.e., $m \gg d^2 \ln d$, we can have $\mathbf{X_S}\mathbf{X_S}^\top \approx \text{diag}(\sum_{i=1}^m |(\mathbf{x}_i)_1|^2, \ldots, \sum_{i=1}^m |(\mathbf{x}_i)_d|^2)$ and $(\mathbf{X_S}\mathbf{X_S}^\top)^{-1} \approx \text{diag}((\sum_{i=1}^m |(\mathbf{x}_i)_1|^2)^{-1}, \ldots, (\sum_{i=1}^m |(\mathbf{x}_i)_d|^2)^{-1})$. Since $\mathbf{w_S} = (\mathbf{X_S}\mathbf{X_S}^\top)^{-1}\mathbf{X_S}\mathbf{y_S}$, the first coordinate value of $\mathbf{w_S}$ is $(\mathbf{w_S})_1 \approx (\sum_{i=1}^m |(\mathbf{x}_i)_1|)/(\sum_{i=1}^m |(\mathbf{x}_i)_1|^2)$.

To more formally state and prove Theorem 1, we provide Theorem 2 to explain the regime of low selection ratio ($m = o(\sqrt{d / \ln d})$) and Theorem 3 for the high selection ratio ($m = \omega(d^2 \ln d)$). To prove the two theorems, we use three lemmas, tail bounds on chi-square and Gaussian distributions, and Gershgorin theorem, which are stated as bellow:

**Lemma 1** (Chi-square tail bound). *If $\mathbf{x} \sim \chi^2(d)$, then $\mathbb{P}(\chi^2(d) \geq d + 2\sqrt{dt} + 2t) \leq e^{-t}$ and $\mathbb{P}(\chi^2(d) \leq d - 2\sqrt{dt}) \leq e^{-t}$.*

**Lemma 2** (Gaussian tail bound). *If $\mathbf{x} \sim \mathcal{N}(0, 1)$, then $\mathbb{P}(|\mathbf{x}| \geq t) \leq e^{\frac{-t^2}{2}}$.*

**Lemma 3** (Gershgorin circle theorem). *Let $A \in \mathbb{C}^{d \times d}$ be a matrix with $(i, j)$-th entry equal to $a_{ij}$. Let $r_i := \sum_{j \neq i} |a_{ij}|$ and $D_i := D_{r_i}(a_{ii})$ be a closed ball centered $a_{ii}$ with radius $r_i$. Then, every eigenvalue of $A$ is contained in $\cup_i D_i$*

Gershgorin circle theorem restricts the eigenvalues of a matrix in a union of disks, whose centers are diagonal elements, and the radius is the sum of off-diagonal elements.

Now, we provide Theorem 2, which will be used to explain why selecting low-scoring (easy) data samples results in a good performance when the subset size $|\mathbf{S}|$ is small.

**Theorem 2** (Sample-deficient regime). *If $m = o\left(\sqrt{d / \ln d}\right)$, then $||(\mathbf{X_S}^\top \mathbf{X_S})^{-1} - \mathbf{I}_m||_2 \leq m\sqrt{\frac{7 \ln n}{2d}}$ with high probability as $d \to \infty$.*

*Proof.* At first, we prove two properties of the high dimensional multivariate normal distribution, which state that the norm of every $\mathbf{x}_i$ is almost equal to 1, and every two independent vectors are almost orthogonal for large enough $d$. For any $1 \leq i \neq j \leq n$, with probability $1 - O(\frac{1}{n})$, we have

$$1 - \sqrt{\frac{7 \ln n}{2d}} \leq \|\mathbf{x}_i\|_2 \leq 1 + \sqrt{\frac{7 \ln n}{2d}}, \quad \text{and} \tag{3}$$

$$|\mathbf{x}_i^\top \mathbf{x}_j| < \sqrt{\frac{7 \ln n}{2d}} \tag{4}$$

The first property (Eq. 3) can be proved by Lemma 1. Let $t = 3 \ln n$ for Lemma 1. Then,

$$\mathcal{P}(\chi^2(d) \geq d + 2\sqrt{3d \ln n} + 6 \ln n) \leq \frac{1}{n^3} \quad \text{and} \quad \mathcal{P}(\chi^2(d) \leq d - 2\sqrt{3d \ln n}) \leq \frac{1}{n^3}$$

Since $2\sqrt{3d \ln n} + 6 \ln n \leq \sqrt{13d \ln n}$ for large enough $d$, with probability $1 - O(\frac{1}{n^3})$ we have

$$1 - \sqrt{\frac{13 \ln n}{d}} \leq \frac{1}{d}\chi^2(d) \leq 1 + \sqrt{\frac{13 \ln n}{d}} \xrightarrow{\text{d} \to \infty} 1 - \sqrt{\frac{7 \ln n}{2d}} \leq \sqrt{\frac{1}{d}\chi^2(d)} \leq 1 + \sqrt{\frac{7 \ln n}{2d}} \tag{5}$$

Since $\mathbf{x}_i \sim \frac{1}{\sqrt{d}}\mathcal{N}(0, \mathbf{I}_d)$ and $\|\mathbf{x}_i\|_2^2 = \frac{1}{d}\chi^2(d)$, for $\forall i \in [n]$, with probability $1 - O(\frac{1}{n^2})$, Eq. 3 follows.

The proof of the second property (Eq. 4) also utilizes Lemma 1. Let $\mathbf{x}_i = \frac{1}{\sqrt{d}}(\mathcal{N}_{i1}, \mathcal{N}_{i2}, \ldots \mathcal{N}_{id})$ and $\mathbf{x}_j = \frac{1}{\sqrt{d}}(\mathcal{N}_{j1}, \mathcal{N}_{j2}, \ldots \mathcal{N}_{jd})$ where $\mathcal{N}_{ik}, \mathcal{N}_{jk}$ are $i.i.d.$ normals $\mathcal{N}(0, 1)$. Then,

$$\mathbf{x}_i^\top \mathbf{x}_j = \frac{1}{d}\sum_{k=1}^{d} \mathcal{N}_{ik}\mathcal{N}_{jk} = \frac{1}{d}\sum_{k=1}^{d} \frac{(\mathcal{N}_{ik} + \mathcal{N}_{jk})^2 - (\mathcal{N}_{ik} - \mathcal{N}_{jk})^2}{4}$$

$$= \frac{1}{2d}\sum_{k=1}^{d}\left[\left(\frac{\mathcal{N}_{ik} + \mathcal{N}_{jk}}{\sqrt{2}}\right)^2 - \left(\frac{\mathcal{N}_{ik} - \mathcal{N}_{jk}}{\sqrt{2}}\right)^2\right] = \frac{1}{2d}\sum_{k=1}^{d}[(\mathcal{N}_k')^2 - (\mathcal{N}_k'')^2]$$

$$= \frac{1}{2d}(\chi_1^2(d) - \chi_2^2(d))$$

where $\mathcal{N}_k'$ and $\mathcal{N}_k''$ are $i.i.d.$ normals, and $\chi_1^2(d)$ and $\chi_2^2(d)$ are $i.i.d$ chi-squares.

As shown in Eq. 5, with probability $1 - O(\frac{1}{n^3})$,

$$1 - \sqrt{\frac{7 \ln n}{2d}} \leq \sqrt{\frac{1}{d}\chi_1^2(d)} \quad \text{and} \quad \sqrt{\frac{1}{d}\chi_2^2(d)} \leq 1 + \sqrt{\frac{7 \ln n}{2d}}. \tag{6}$$

Thus, we have

$$\left|\frac{1}{2d}(\chi_1^2(d) - \chi_2^2(d))\right| \leq \sqrt{\frac{7 \ln n}{2d}}. \tag{7}$$

By applying a union bound, for $\forall i \neq j \in [n]$, with probability $1 - O(\frac{1}{n})$, we have $|\mathbf{x}_i^\top \mathbf{x}_j| \leq \sqrt{\frac{7 \ln n}{2d}}$. From Eq. 3 and Eq. 4, we obtain that $\|\mathbf{X}_S^\top \mathbf{X}_S - \mathbf{I}_m\|_F^2 \leq m^2\left(\frac{7 \ln n}{2d}\right)$.

Let $\mathbf{A} = \mathbf{X}_S^\top \mathbf{X}_S$, then we derive the bound on $\|\mathbf{I} - \mathbf{A}^{-1}\|_2$ from the bounds of $\|\mathbf{I} - \mathbf{A}\|_2$ and $\|\mathbf{A}^{-1}\|_2$. First, note that

$$\|\mathbf{I} - \mathbf{A}\|_2 \leq \|\mathbf{I} - \mathbf{A}\|_F \leq m\sqrt{\frac{7 \ln n}{2d}} \quad \text{and} \quad m\sqrt{\frac{7 \ln n}{2d}} \to 0 \text{ as } d \to \infty$$

since $m = o(\sqrt{d/\ln d})$ and $n = poly(d)$. Moreover, we have

$$\|\mathbf{A}^{-1}\|_2 = \|(\mathbf{I} - (\mathbf{I} - \mathbf{A}))^{-1}\|_2 = \|\mathbf{I} + (\mathbf{I} - \mathbf{A}) + (\mathbf{I} - \mathbf{A})^2 + \ldots\|_2$$

$$\leq \|\mathbf{I}\|_2 + \|(\mathbf{I} - \mathbf{A})\|_2 + \|(\mathbf{I} - \mathbf{A})^2\|_2 + \cdots \leq 1 + \sum_{k=1}^{\infty}\left(m\sqrt{\frac{7 \ln n}{2d}}\right)^k \leq 2.$$

Finally, we have

$$\|\mathbf{I} - \mathbf{A}^{-1}\|_2 \leq \|\mathbf{A}^{-1}\|_2 \|\mathbf{I} - \mathbf{A}\|_2 \leq m\sqrt{\frac{7\ln n}{2d}}.$$

$\square$

We next provide Theorem 3, which explains why selecting high-scoring (difficult) data samples results in a good for performance when the subset size $|\mathbf{S}|$ is large ($m = \omega(d^2 \ln d)$). Assume that we select the subset $\mathbf{X_S}$ by observing the first element of each data, $(\mathbf{x}_i)_1$. Suppose that we select the data samples whose first elements are $\frac{a_1}{\sqrt{d}}, \frac{a_2}{\sqrt{d}}, \dots \frac{a_m}{\sqrt{d}}$ where $a_i \in \Theta(1)$, and let $a := \frac{\sum_{i=1}^m a_i^2}{m}$. The elements of the other coordinates are independent normals, i.e., $(\mathbf{x}_i)_k \sim \mathcal{N}(0,1)$ for $k \geq 2$. We will prove that $(\frac{d}{m}\mathbf{X_S}\mathbf{X_S}^\top)^{-1}$ can be approximated by a diagonal matrix of which the first element is equal to $\frac{1}{a}$ and other elements are equal to 1.

**Theorem 3** (Sample-sufficient regime). *Let* $\mathbf{B} = \mathrm{diag}(a, 1, 1, \dots 1) \in \mathbb{R}^{d \times d}$. *If* $m = \omega(d^2 \ln d)$, *then* $\|(\frac{d}{m}\mathbf{X_S}\mathbf{X_S}^\top)^{-1} - B^{-1}\|_2 \leq c'd^2\frac{\ln d}{m}$ *for some constant* $c' > 0$ *with high probability as* $d \to \infty$.

*Proof.* The elements of $\frac{d}{m}\mathbf{X_S}\mathbf{X_S}^\top$ are expressed as follows, where $k \neq l \in [d]\setminus\{1\}$.

$$\frac{d}{m}(\mathbf{X_S}\mathbf{X_S}^\top)_{11} = \frac{1}{m}\sum_{i=1}^m a_i^2 = a$$

$$\frac{d}{m}(\mathbf{X_S}\mathbf{X_S}^\top)_{k1} = \frac{1}{m}\sum_{i=1}^m a_i\mathcal{N}_i = \mathcal{N}\left(0, \frac{\sum_{i=1}^m a_i^2}{m^2}\right) = \mathcal{N}\left(0, \frac{a}{m}\right)$$

$$\frac{d}{m}(\mathbf{X_S}\mathbf{X_S}^\top)_{kk} = \frac{d}{m}\sum_{i=1}^m (\mathbf{x}_i)_k^2 = \frac{1}{m}\mathcal{N}_{ik}^2 = \frac{1}{m}\chi^2(m)$$

$$\frac{d}{m}(\mathbf{X_S}\mathbf{X_S}^\top)_{kl} = \frac{d}{m}\sum_{i=1}^m (\mathbf{x}_i)_k(\mathbf{x}_i)_l = \frac{1}{m}\sum_{i=1}^m \mathcal{N}_{ik}\mathcal{N}_{il}$$

By Gaussian tail bound (Lemma 2), if $\mathbf{x} \sim \mathcal{N}(0, a/m)$, then we have

$$\mathbb{P}(|\mathbf{x}| \geq 2\sqrt{a\ln d/m}) \leq 1/d^2.$$

Note that $\frac{1}{m}\chi^2(m) = \frac{1}{m}\|\mathbf{x}\|_2^2$ for $\mathbf{x} \sim \mathcal{N}(0, \mathbf{I}_m)$. By Lemma 1 we have a result similar to Eq. 4,

$$\mathbb{P}(|\|\mathbf{x}\|_2 - 1| \geq \sqrt{7\ln d/2m}) \leq 1/d^2.$$

For $\mathcal{N}_{ik}\mathcal{N}_{il}$, by applying the result of Eq. 4, we can also show that

$$\mathbb{P}(\|\mathbf{x}\|_2 \geq \sqrt{7\ln d/2m}) \leq 1/d^3.$$

Combining the above three bounds, we obtain that for $\forall k \neq l \in [d]$, with probability $1 - O(\frac{1}{d})$,

$$\left|\frac{d}{m}(\mathbf{X_S}\mathbf{X_S}^\top)_{k1}\right| \leq 2\sqrt{\frac{a\ln d}{m}}, \quad \left|\frac{d}{m}(\mathbf{X_S}\mathbf{X_S}^\top)_{kk} - 1\right| \leq \sqrt{\frac{7\ln d}{2m}}, \text{ and } \left|\frac{d}{m}(\mathbf{X_S}\mathbf{X_S}^\top)_{kl}\right| \leq \sqrt{\frac{7\ln d}{2m}}.$$

Thus, we obtain $\|\frac{d}{m}(\mathbf{X_S}\mathbf{X_S}^\top) - \mathbf{B}\|_F \leq cd^2\frac{\ln d}{m}$ for some constant $c > 0$, with probability $1 - O(\frac{1}{d})$. Let $\mathbf{A} = \frac{d}{m}(\mathbf{X_S}\mathbf{X_S}^\top)$, then

$$\|\mathbf{A}^{-1} - \mathbf{B}^{-1}\|_2 \leq \|\mathbf{A}^{-1}\|_2\|\mathbf{I} - \mathbf{A}\mathbf{B}^{-1}\|_2 \leq \|\mathbf{A}^{-1}\|_2\|\mathbf{B} - \mathbf{A}\|_2\|\mathbf{B}^{-1}\|_2$$

$$\leq \|\mathbf{A}^{-1}\|_2\|\mathbf{B} - \mathbf{A}\|_F\|\mathbf{B}^{-1}\|_2 \leq \|\mathbf{A}^{-1}\|_2 \, cd^2\frac{\ln d}{m} \cdot 1.$$

It is remaining to prove that $\|\mathbf{A}^{-1}\|_2$ is bounded. The eigenvalues of $\mathbf{A} = \frac{d}{m}(\mathbf{X_S}\mathbf{X_S}^\top)$ are almost equal to the diagonal elements by utilizing Gershgorin circle theorem. Since $m \in \omega(d^2 \ln d)$,

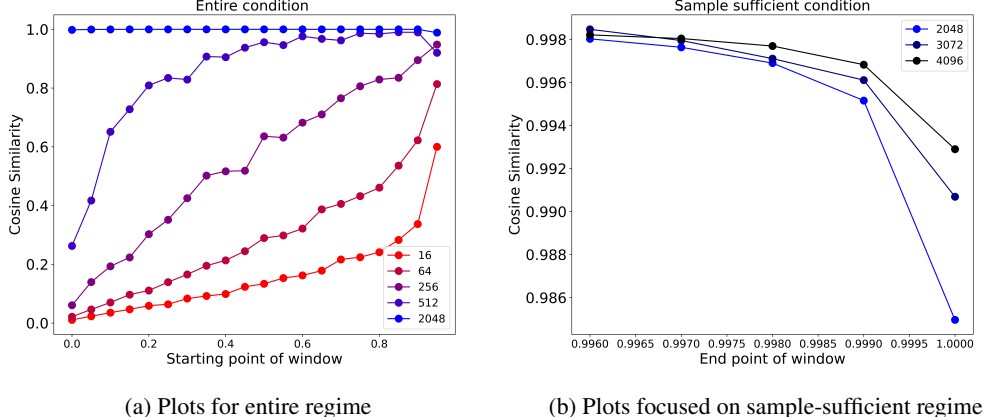

(a) Plots for entire regime

(b) Plots focused on sample-sufficient regime

Figure 5: Results of window sliding experiment at the setting of theoretical analysis. In our setting, the dimension $d$ is 256, the number of full dataset $n$ is $256,000$, and the subset size $m$ are selected among $16, 64, 256, 512, 2048, 3072,$ and $4096$. Left figure covers the entire regime ($m = 16, 64, 256, 512,$ and $2048$), while the right figure focuses on sample-sufficient regime.

$$r_1 = \sum_{j \neq i} |a_{1j}| < 2d\sqrt{\frac{a \ln d}{m}} \ll a \quad \text{and} \quad r_k = \sum_{j \neq i} |a_{kj}| < d\sqrt{\frac{7 \ln d}{2m}} \ll 1.$$

Therefore, every eigenvalues of $\mathbf{A}$ are close to either $a$ or $1$ with probability $1 - O(\frac{1}{d})$. Thus, $\|\mathbf{A}^{-1}\|_2$ is bounded above by $\max(\frac{1}{a}, 1)$ plus some some constant, which shows that $\|\mathbf{A}^{-1}\|_2$ is bounded. Therefore, we have

$$\|\mathbf{A}^{-1} - \mathbf{B}^{-1}\|_2 \leq \|\mathbf{A}^{-1}\|_2 \, cd^2 \frac{\ln d}{m} \leq c'd^2 \frac{\ln d}{m} \quad \text{for some constant } c' > 0.$$

$\square$

### A.3 TOY EXPERIMENT

To validate our theoretical analysis, we conduct a window sliding experiment similar to the one in Sec. 4, at the setting of the theoretical analysis in Sec. A.2, while varying the subset sizes and the starting points of the window subsets at $d = 256$ and $n = 256,000$. The results are shown in Fig. 5. Fig. 5a shows the plot for both the sample-deficient and sufficient regimes, including $m = 16, 64, 256, 512, 2048$, while Fig. 5b shows focused plots for sample sufficient regime where $m = 2048, 3072, 4096$. The x-axis in Fig. 5a is the starting point of the window subset, which identifies the ranking of the hardest sample in the window subset, while that in Fig. 5b is the end point of the window subset, which identifies the ranking of the easiest sample in the window subset. The y-axis is the cosine similarity between $\mathbf{w}$ and $\hat{e}_1 = (1, 0, \ldots, 0)$, where $\mathbf{w}$ is the solution of the regression problem, and $\hat{e}_1$ is the unit vector with its first coordinate equal to 1, which is the true decision boundary. A higher cosine similarity implies a better solution. Red lines show the results when subset size $m$ is smaller, and the blue or black lines show the result of larger subset sizes.

In the sample-deficient regime where $m \leq d$ (red lines), the cosine similarity increases as the starting point of the window increases, meaning that it is better to use easy samples to learn the linear classifier. On the other hand, in the sample-sufficient regime where $m > d$ (blue and black lines), the cosine similarity is larger for windows having a smaller end point, meaning that it is better to include difficulty samples to learn a better classifier. This result coincides with the theoretical analysis, which claims that the inclusion of easier (harder) data samples results in a better solution for a smaller (larger) subset size, respectively. As we conjectured at Sec. 3.1, the transition of a desirable selection strategy occurs near $m = \Theta(d)$.

# B    DISCUSSIONS ON USING KERNEL RIDGE REGRESSION AS A PROXY

In Algorithm 1, we use the kernel ridge regression as a proxy for training neural networks to evaluate the performance of window subsets. In this section, we provide some theoretical rationale behind the use of the kernel ridge regression, and also compare it with another proxy using a smaller network, motivated from Selection Via Proxy (SVP) (Coleman et al., 2020).

## B.1    KERNEL RIDGE REGRESSION AS A PROXY OF NEURAL NETWORK TRAINING

Our use of the kernel ridge regression as a proxy for training neural networks can be partly explained by the recent progress in theoretical understanding of training neural networks using kernel methods. In particular, some recent works (Neal, 2012; Lee et al., 2018; Jacot et al., 2018; Arora et al., 2019) have shown that training and generalization of neural networks can be approximated by two associated kernel matrices: the Conjugate Kernel (CK) and Neural Tangent Kernel (NTK). The Conjugate Kernel is defined by the gram matrix of the derived features produced by the final hidden layer of the network, while NTK is the gram matrix of the Jacobian of in-sample predictions with respect to the network weights. These two kernels also have fundamental relations in terms of their eigenvalue distributions as analyzed in Fan & Wang (2020). Our proxy task is motivated by the observation that the kernel regression with these model-related kernels can provide a good approximation to the original model (under some assumptions such as enough width, random initialization, and small enough learning rate, etc.). As an example, the work by Arora et al. (2019) provides the following theorem, which connects the training of a neural network with kernel ridge regression using NTK.

**Theorem 4** (Informal version of Arora et al. (2019)). *Consider a fully connected neural network with sufficiently large width $d_1 = d_2 = \ldots d_L$ where $d_l$ is number of nodes in lth layer. Given a training dataset $\{(\mathbf{x}_i, y_i)\}_{i=1}^n \subset \mathbb{R}^d \times \mathbb{R}$ with normalized inputs $\|\mathbf{x}_i\|_2 = 1$, the network is trained by gradient descent with a sufficiently small learning rate to minimize the square-loss $\sum_{i=1}^n (f_{nn}(\mathbf{x}_i) - y_i)^2$. Let $f_{nn}$ be the trained network. With a kernel function of the network $K(\cdot, \cdot)$, NTK of training data $\mathbf{H} \in \mathbb{R}^{n \times n}$ is defined by $\mathbf{H}_{ij} = K(\mathbf{x}_i, \mathbf{x}_j)$. And, for a test data $\mathbf{x}_{te}$, the kernel between the test data and the training dataset $\mathbf{X} = [\mathbf{x}_1, \mathbf{x}_2, \ldots \mathbf{x}_n] \in \mathbb{R}^{d \times n}$ is defined by $\mathbf{K}(\mathbf{x}_{te}, \mathbf{X}) \in \mathbb{R}^n$ where $\mathbf{K}(\mathbf{x}_{te}, \mathbf{X})_i = K(\mathbf{x}_{te}, \mathbf{x}_i)$. Let $f_{ntk}(\mathbf{x}_{te}) = (\mathbf{K}(\mathbf{x}_{te}, \mathbf{X}))^\top \mathbf{H}^{-1}\mathbf{y}$. Then,*

$$|f_{nn}(\mathbf{x}) - f_{ntk}(\mathbf{x})| \leq \epsilon$$

Theorem 4 justifies that the kernel regression with NTK can provide a good approximation to the neural network training under the specified assumptions. However, calculating the NTK for the entire neural network requires high computational cost as it involves computing the Jacobian with respect to the network weights.

To address such a problem, Conjugate Kernel is often considered as a promising replacement of NTK. For example, the work by Zhou et al. (2022) utilizes the kernel ridge regression based on Conjugate Kernel for dataset distillation. We also use the Conjugate Kernel in our kernel ridge regression (Eq. 1), by defining the kernel matrix as $\mathbf{X}_\mathbf{S}^\top \mathbf{X}_\mathbf{S}$ where $\mathbf{X}_\mathbf{S} = [\mathbf{f}_1, \ldots, \mathbf{f}_m]$ is composed of features produced by the exact target network of our consideration (ResNet18 for CIFAR-10 and ResNet50 for CIFAR-100/ImageNet). By considering the features from the target network, we can obtain the (approximate) network predictions that are linear in these derived features. In detail, the output of the CK-regression for the test example $\mathbf{x}_{te}$ can be written as $f_{ntk}(\mathbf{x}_{te}) = \mathbf{x}_{te}^\top \mathbf{X}_\mathbf{S}(\mathbf{X}_\mathbf{S}^\top \mathbf{X}_\mathbf{S})^{-1}\mathbf{y}_\mathbf{S} = \mathbf{x}_{te}^\top \mathbf{w}$ where $\mathbf{w} = \mathbf{X}_\mathbf{S}(\mathbf{X}_\mathbf{S}^\top \mathbf{X}_\mathbf{S})^{-1}\mathbf{y}$ in Eq. 2 for $\lambda = 0$.

Of course, this kernel approximation of the neural network models, which assumes a fixed feature extractor, does not exactly match our situation where the selected subset not only affects the linear classifier but also the feature extractor itself during the training. However, this is still a good proxy that can reflect the network architecture of our interest in a computationally-efficient manner. Also, our analysis in Table 2 shows that this proxy finds the best window subset that aligns well with the result from the actual training of the full model.

## B.2    SELECTION VIA PROXY FOR EVALUATING THE WINDOW SUBSETS

Selection Via Proxy (SVP) (Coleman et al., 2020) utilizes a small model for evaluating the importance of data instances, and then apply a selection method (e.g. difficulty-score) to choose a subset

Table 5: Test accuracy of the models trained by window subsets of CIFAR-10 selected by KRR (kernel ridge regression, ours) and SVP (selection via proxy). We observe that SVP tends to select easier samples (windows with larger index) compared to the oracle window subset possibly due to the limited capability of the simple network used in the proxy task.

| Selection methods | Selection ratio | 1% | 5% | 10% | 20% | 30% | 40% | 50% | 75% | 90% |
|---|---|---|---|---|---|---|---|---|---|---|
| KRR (ours) | Test accuracy | 63.14 | 81.31 | 88.05 | 91.29 | 93.17 | 94.38 | 94.93 | 95.20 | 95.22 |
| | Window index | 90% | 70% | 35% | 25% | 15% | 5% | 0% | 0% | 0% |
| SVP | Test accuracy | 65.73 | 81.78 | 86.40 | 90.18 | 91.63 | 92.50 | 93.07 | 94.96 | 95.22 |
| | Window index | 80% | 60% | 55% | 40% | 30% | 5% | 0% | 0% | 0% |
| Oracle window | Test accuracy | 65.73 | 83.03 | 88.05 | 91.69 | 93.35 | 94.38 | 94.93 | 95.20 | 95.22 |
| | Window index | 80% | 50% | 35% | 20% | 10% | 5% | 0% | 0% | 0% |

that will be evaluated on (more complex) target models. To evaluate the efficiency of this approach, we conduct an experiment of finding the best window subset using a small ConvNet (Zhao et al., 2021) model on the CIFAR-10 dataset. The selected subset is then evaluated on ResNet18. In Table 5, we summarize this result (SVP) compared to our original proxy (KRR, Kernel Ridge Regression) and the oracle window using the target model. We can observe that SVP tends to select easier samples (windows with larger index) compared to the oracle window over the selection ratios 5% to 90%, which results in performance loss especially in high selection ratio regimes. Our proxy, on the other hand, exactly matches the oracle window performance at selection ratios of 10% to 90%. We conjecture that the tendency that SVP selects an easier subset is attributed to the limited capability of the simple network used in the proxy task. Furthermore, solving our proxy task takes only about 1/15-1/250 (varying depending on the subset size) of the computational time compared to SVP, which requires training of the small network (ConvNet) for all considered subsets.

## C  IMPLEMENTATION DETAILS

### C.1  BASELINE DETAILS

We benchmark our BWS algorithm against six state-of-the-art methods, Forgetting score (Toneva et al., 2019), EL2N score (Paul et al., 2021), AdaCore (Pooladzandi et al., 2022), LCMat (Shin et al., 2023), Moderate DS (Xia et al., 2023), and CCS (Zheng et al., 2023).

In Forgetting and EL2N scores, scores are derived by averaging results of five independent training using the full CIFAR-10/100 dataset. Specifically, Forgetting scores are obtained at the 200th epoch (full training), while EL2N scores are captured at the 20th epoch. For our ImageNet experiments, pre-calculated Forgetting and EL2N scores are sourced from https://github.com/rgeirhos/dataset-pruning-metrics Sorscher et al. (2022).

In the AdaCore methodology, subset selection is computed only once, at the 10th epoch, to ensure a fair comparison. The LCMat approach mirrors this protocol, with selection also occurring at the 10th epoch. Both AdaCore and LCMat implementations are sourced from the LCMat repository. For Moderate DS, models are trained using the full dataset, and the individual data features are extracted from the models. These features are defined as the outputs of the penultimate layer, with dimensions being 512 for ResNet18 and 2048 for ResNet50. The CCS algorithm employs the aforementioned Forgetting score. Within the CCS approach, we consistently set the hyperparameter $\beta$ to zero across all data selection ratios.

All computational tasks utilized consistent network architectures, as detailed in Section 5: ResNet18 for CIFAR-10 and ResNet-50 for both CIFAR-100 and the ImageNet dataset. Additional experimental specifications related with learning algorithm are reported in Table 6 of Appendix §C.2.

Details of the baselines are summarized below:

- Error L2-Norm (EL2N): The EL2N score of data $(\mathbf{x}_i, y_i)$ is defined as $\mathbb{E}[\|f(\mathbf{W}(t), \mathbf{x}_i) - y_i\|_2]$, where $f(\mathbf{W}(t), \mathbf{x})$ is the output of the neural network for the sample $(\mathbf{x}, y)$ at the $t$-th epoch.

- Forgetting score: The Forgetting score is defined as the number of times during training (until epoch $T$) that the decision of the sample switches from a correct one to an incorrect one: Forgetting$(\mathbf{x}_i, y_i)$ is defined as

$$\sum_{t=2}^{T} \mathbb{1}\{\arg\max f(\mathbf{W}(t-1), \mathbf{x}_i) = y_i\}(1 - \mathbb{1}\{\arg\max f(\mathbf{W}(t), \mathbf{x}_i) = y_i\}). \quad (8)$$

- AdaCore: Adaptive Second order Coresets (AdaCore) is an algorithm that solves the optimization problem, which finds a subset that imitates the full gradient preconditioned with the Hessian matrix:

$$S^* \in \arg\min_{S \subset V} \sum_{i \in V} \min_{j \in S} \|\mathbf{H}_i(w_t)^{-1}\mathbf{g}_i(w_t) - \mathbf{H}_j(w_t)^{-1}\mathbf{g}_j(w_t)\|, \text{ s.t. } |S| \le r \quad (9)$$

where $\mathbf{g}_i(w_t) = \nabla l(w_t, (\mathbf{x}_i, y_i))$ and $\mathbf{H}_i(w_t) = \nabla^2 l(w_t, (\mathbf{x}_i, y_i))$ represent the gradient and Hessian of the loss function for the data point $(\mathbf{x}_i, y_i)$ using the model parameter $w_t$ at the $t$-th epoch of training, respectively. Let $V$ represent the full dataset and $S$ be the coreset of size $r$. In the AdaCore method, when employing the cross entropy loss with a softmax layer as the final layer, the gradient $\mathbf{g}_i(w_t)$ is approximated by $p_i - y_i$, where $p_i$ is the softmax output for the data point $(\mathbf{x}_i, y_i)$. Moreover, to reduce computational complexity, the Hessian $\mathbf{H}_i(w_t)$ is approximated using only its diagonal.

- LCMat: Loss-Curvature Matching (LCMat) is an algorithm that solves the optimization problem, which finds a subset that matches the loss curvature of full dataset. Because of intractability of utilizing the loss curvature, the suggested alternative optimization problem is as follows:

$$S^* \in \arg\min_{S \subset V} \sum_{i \in V} \min_{j \in S} \|\mathbf{g}_i(w_t) - \mathbf{g}_j(w_t)\| + \frac{1}{2}\rho \sum_{k \in \mathcal{K}} |\lambda_{i,k} - \lambda_{j,k}|, \text{ s.t. } |S| \le r \quad (10)$$

where $\mathbf{g}_i(w_t) = \nabla l(w, (\mathbf{x}_i, y_i))$ and $\lambda_{i,k} = \mathbf{H}_i(w_t)_{kk} = \nabla^2_{kk} l(w, (\mathbf{x}_i, y_i))$ denote the gradient and the $k$-th diagonal element of the Hessian of the loss function for the data point $(\mathbf{x}_i, y_i)$ with the model parameter $w_t$ at the $t$-th epoch of training, respectively. Let $V$ represent the full dataset, $S$ the coreset with size $r$ and $W$ the model parameter space. $\mathcal{K} = \arg\max_{|\mathcal{K}|=K} \sum_{j \in \mathcal{K}} \text{Var}_i(\lambda_{i,k})$ is a set of indices for $K$ sub-dimensions on $w$, where the dimension variance is high. In LCMat, when employing the cross entropy loss with a softmax layer as the final layer, the gradient $\mathbf{g}_i(w)$ is approximated by $p_i - y_i$, where $p_i$ is the softmax output for the data point $(\mathbf{x}_i, y_i)$.

- Moderate DS: For each class of a given dataset, Moderate Coreset calculates distance between feature and the center of the class, which is defined as $d_i = ||\mathbf{f}_i - \frac{\sum_{j \in S} \mathbf{f}_j}{|S|}||_2$ where $S$ is the set of features whose label is the same as $f_i$. Then, data points with distances closest to the distance-median($median(\{d_i\}_{i \in S})$) are selected.

- CCS: Coverage-Centric Coreset Selection is an algorithm based on difficulty-based score, which considers overall data coverage upon a distribution as well as important data. CCS prunes $\beta\%$ hardest data first and splits the remained data into $k$ subsets $\{\mathbf{B}_i\}_{i=1}^{k}$ based on evenly divided score ranges $\{\mathbf{R}_i\}_{i=1}^{k}$. Then, CCS selects the same number of samples from each score range to make the score distribution of the selected samples uniform.

## C.2 EXPERIMENT DETAILS

**Data pruning experiment**   We construct experiments with three public datasets, CIFAR-10/100 and ImageNet by training ResNet networks (He et al., 2016) of different depths. ResNet18 is used for CIFAR-10 and ResNet50 is used for CIFAR-100 and ImageNet dataset. Implementation of the ResNet is based on the ResNet network in torchvision (Paszke et al., 2019). Since CIFAR-10/100 images are smaller than ImageNet images, we replace the front parts of the ResNet (convolution layer with 7x7 kernel and 2x2 stride, max pooling layer with 3x3 kernel and 2x2 stride) with a single convolution layer with 3x3 kernel and 1x1 stride for small size image. The details on hyperparameters and optimization methods used in training are summarized in the Table 6.

Table 6: Details for the experiments used in the training of the dataset.

|  | CIFAR-10 | CIFAR-100 | ImageNet |
| --- | --- | --- | --- |
| Architecture | ResNet18 | ResNet50 | ResNet50 |
| Batch size | 128 | 128 | 256 |
| Epochs | 200 | 200 | 90 |
| Initial Learning Rate | 0.05 | 0.1 | 0.1 |
| Weight decay | 5e-4 | 5e-4 | 1e-4 |
| Learning Rate Scheduler | Cosine annealing scheduler | | Step scheduler |
| Optimizer | SGD with momentum 0.9 | | |
| Data Augmentation | Random Zero Padded Cropping (4 pixels) Random left-right flipping (probability 0.5) Normalize by dataset's mean, variance | | Random Resized Cropping |

Our experiments report averaged results from three runs on CIFAR-10/100 and two on ImageNet, with shaded regions representing standard deviations. Networks are trained on datasets curated based on specific selection ratios and methods. For CIFAR-10/100, we determine the number of iterations to be the same as the number of iterations required to process the entire dataset within one epoch and maintain this count across experiments. Crucially, our data selection ensures equal selection from each class by preserving the portion data in each class.

**Cross-architecture robustness** We conduct cross-architecture experiments on the CIFAR-10 datasets, training two distinct networks: a simple CNN and a Vision Transformer (ViT) pretrained on the ImageNet dataset.

For the simple CNN, we design an architecture comprising three convolutional layers with a $3 \times 3$ kernel and $1 \times 1$ stride (channels: 64, 128, 256). This is paired with two max-pooling layers with a $2 \times 2$ kernel. The convolutional layers are interspersed with these max-pooling layers. Following the convolutional layers, the network is connected to two fully connected layers (channels: 128, 256). Each convolutional layer is equipped with a batch normalization layer followed by a non-linear ReLU activation layer. We set the initial learning rate to 0.05 and weight decay to 1e-4. Other details are the same as Sec. C.2 of CIFAR-10 case. For the ViT, we adhere to the implementation specifications as detailed in Dosovitskiy et al. (2021). We obtain a ViT model pretrained on the ImageNet dataset using the timm module in PyTorch, which we subsequently fine-tune on the CIFAR-10 dataset for 10 epochs. For fine-tuning a model pre-trained on ImageNet to adapt to the CIFAR-10 dataset, we resized the data to fit the 224x224 pixel dimensions. We set the initial learning rate to 1e-4 and weight decay to 1e-4. We do not use a learning rate scheduler. Other details are the same as those in Sec. C.2 of CIFAR-10 case.

Within our algorithm, BWS, we utilize a forgetting score sourced from the ResNet18 architecture. Furthermore, we establish a feature extractor using the simple CNN and ViT architectures individually. For the CNN architecture, we execute training for 20 epochs, while for the ViT setup, we fine-tune for 3 epochs.

We report the averaged results from three independent runs on the two networks, with the shaded regions indicating the standard deviations. Like in previous experiments, data are selected to ensure a balanced portion of each class, preserving the original class ratios within the CIFAR-10 dataset.

**Robustness to label noise** We create noisy CIFAR10 dataset with 20% symmetric label noise. We compute the EL2N score of this noisy dataset using ResNet18 and average the results over 5 independent runs. We utilize EL2N score since this score has a better noise discrimination ability, and is cheaper in computational cost compared to Forgetting score in calculating the new set of EL2N scores for the noisy dataset. When we apply Alg. 1 with the noisy CIFAR-10 dataset ordered by the EL2N scores, we set $t$ of Alg. 1 to 5% (or 2500), and calculate the classification accuracy of $\mathbf{w_S}$ by using the low-scoring 50% samples, i.e., by $\frac{1}{n} \sum_{i=\frac{n}{2}}^{n} \mathbb{1}(\arg\max_c(\mathbf{w_S^\top x}_i)_c = y_i)$ instead of by $\frac{1}{n} \sum_{i=1}^{n} \mathbb{1}(\arg\max_c(\mathbf{w_S^\top x}_i)_c = y_i)$ to avoid including noisy samples in quality evaluation of window subsets.

**Ablation on different window selection methods**    The formal definitions of Gradient difference and Gradient similarity are as follows:

• Gradient difference: minimizing the difference between the gradients of the full training dataset ($V$) and window subset ($S$).

$$\text{Gradient Difference}(V, S) = \left\| \frac{\sum_{i \in V} \nabla f_{\mathbf{w}}(\mathbf{x}_i)}{|V|} - \frac{\sum_{i \in S} \nabla f_{\mathbf{w}}(\mathbf{x}_i)}{|S|} \right\|_2 \qquad (11)$$

• Gradient similarity: maximizing the cosine similarity between the gradients of the full training dataset ($V$) and window subset ($S$).

$$\text{Gradient Similarity}(V, S) = \frac{\sum_{i \in V} \nabla f_{\mathbf{w}}(\mathbf{x}_i)}{\left\| \sum_{i \in V} \nabla f_{\mathbf{w}}(\mathbf{x}_i) \right\|_2} \cdot \frac{\sum_{i \in S} \nabla f_{\mathbf{w}}(\mathbf{x}_i)}{\left\| \sum_{i \in S} \nabla f_{\mathbf{w}}(\mathbf{x}_i) \right\|_2} \qquad (12)$$

### C.3   Computational cost

Table 7: Time cost (in seconds) to compute sample scores of the dataset.

| Selection ratio | | 1% | 5% | 10% | 20% | 30% | 40% | 50% | 75% | 90% |
|---|---|---|---|---|---|---|---|---|---|---|
| | BWS (Ours) | 4.3 | 4.8 | 7.2 | 8.5 | 9.7 | 10.4 | 10.4 | 9.0 | 6.3 |
| CIFAR-10 | LCMat | 520 | 1197 | 2260 | 4213 | 5800 | 7173 | 8450 | 10320 | 10631 |
| | AdaCore | 224 | 699 | 1256 | 2273 | 3186 | 3977 | 4649 | 5637 | 5839 |
| | BWS (Ours) | 14.6 | 55.6 | 57.3 | 62.2 | 64.7 | 64.0 | 61.7 | 45.7 | 30.2 |
| CIFAR-100 | LCMat | 1465 | 1468 | 1471 | 1478 | 1483 | 1489 | 1493 | 1501 | 1504 |
| | AdaCore | 1295 | 1300 | 1304 | 1309 | 1315 | 1320 | 1324 | 1331 | 1334 |
| | BWS (Ours) | 1423 | 2910 | 3941 | 6141 | 7590 | 8616 | 9029 | 10118 | 6499 |
| ImageNet | LCMat | 238451 | 239027 | 239694 | 240934 | 242003 | 242864 | 243594 | 244854 | 245245 |
| | AdaCore | 213733 | 214309 | 214963 | 216181 | 217067 | 217866 | 218521 | 219593 | 219924 |
| GPU | | Nvidia A100 40GB | | | | | | | | |

We compare the computational cost of our BWS algorithm, as detailed in Algorithm 1, with other optimization-based coreset selection baselines, namely LCMat and AdaCore. We assume that the sample scores, used for sorting, is readily available and that the feature extractor is also pre-provided for both ours and optimization-based methods. We report and compare the time taken to select the subset for each algorithm.

In Table 7, we detail the time required to select subsets from various datasets using the different methods. Clearly, our method outperforms optimization-based techniques in terms of time cost for subset selection. As we have previously described in Sec. 4, our strategy, which selects the best window subset from a continuous interval of samples sorted by their scores, greatly reduces the search space compared to the general optimization techniques, leading to improved efficiency.

## D   Detailed Review of Related Works

In this section, we provide additional related works on data subset selection that employ various different perspectives.

When there is no validation set, some score-based selection methods, such as EL2N (Paul et al., 2021) and Forgetting (Toneva et al., 2019), suffer from performance degradation when the dataset includes label-noise samples, since these methods often assign high scores to label-noisy samples, as label-noise samples are inherently hard to learn. Some recent methods adopt more cautious measures to distinguish hard-to-learn but clean-label samples, known to be valuable to enhance the generalization ability of neural networks, from label-noise samples. For instance, Cartography (Swayamdipta et al., 2020) utilizes two measures, confidence mean and confidence variance of data sample, to distinguish hard-to-learn samples from mere label-noise samples. Second-Split Forgetting (Maini et al., 2022) achieves this goal by observing learning time and forgetting time and splitting the training dataset. AUM (Pleiss et al., 2020) observes logit value of a given label and the next largest logit, and use the gap to separate noisy data samples and ambiguous data samples.

Another important perspective that has recently explored in data subset selection is the computational overhead in quantifying the data value and the model-dependent issue on data valuation. There are several recent attempts to valuate data without training of a neural network. CG-score (Ki et al., 2023) evaluates data instances without model training by calculating the analytical gap in generalization errors when an instance is held out. LAVA(Just et al. (2023)) evaluate the value of each data instance without a model training by using a proxy function, which is class-wise Wasserstein distance between training and validation set, for the validation performance. DAVINZ (Wu et al., 2022) utilizes the Neural Tangent Kernel (NTK) of a network at initialization for calculating the contribution of each data instance to domain-aware generalization error bound.

In optimization-based selection, many works observe the effect of data on the model during training. MaxMargin (Har-Peled et al., 2007), IWeS (Citovsky et al., 2023), and Selection-via-proxy (Coleman et al., 2020) observe the confidence of the model to identify uncertain data during optimization. Maximum Margin Coresets (Har-Peled et al., 2007) selects data with the smallest margin in an SVM setting, IWeS (Citovsky et al., 2023) selects examples using importance sampling with a sampling probability based on the confidences of two models, and Selection-via-proxy (Coleman et al., 2020) applies confidence-based methods to a small proxy model to perform data selection. Work by (Das et al., 2021) solves a convex linear programming problem to find high-value data that contributes much to the loss and optimization, and Glister (Killamsetty et al., 2021b) finds data that contributes significantly to the loss during the training of a neural network. GradMatch (Killamsetty et al., 2021a) finds a subset whose gradients are matched with the gradient of the full dataset.

Additionally, in active learning, which selects data to be labeled for semi-supervised learning, Neural-Preconditioning (Kong et al., 2022) obtains the label of data that dominates the eigenspace of the NTK, and the work by (Culotta & McCallum, 2005) obtains the label of the least confident data to reduce labeling costs.

(Har-Peled et al., 2007; Citovsky et al., 2023; Coleman et al., 2020) observe confidence of model to identify uncertain data during the optimization, and (Das et al., 2021; Killamsetty et al., 2021b) find data that contributes much to the loss.

# E  SLIDING WINDOW EXPERIMENT IN CIFAR-100 AND IMAGENET DATASETS

In this section, we report the results of the sliding window experiments, where we change the starting point of the window subsets and compare the corresponding performances in terms of 1) the test accuracy on the model trained by each subset and 2) the accuracy of a proxy task using kernel ridge regression on training dataset. We report the results on CIFAR-100 and ImageNet.

Table 8: Comparison of window subsets of CIFAR-100 in terms of their 1) test accuracy, measured on models trained with the window subsets (top rows) and 2) accuracy of kernel ridge regression on the training dataset (bottom rows). The best performing windows align well between the two measures.

| Ratio | Starting point | 0% | 5% | 10% | 15% | 20% | 25% | 30% | 35% | 40% | 45% | 50% | 55% | 60% | 65% | 70% | 75% | 80% | 85% | 90% |
|---|---|---|---|---|---|---|---|---|---|---|---|---|---|---|---|---|---|---|---|---|
| 10% | Test Acc | 32.63 | 31.20 | 33.34 | 31.71 | 34.77 | 36.05 | 34.00 | 35.52 | 35.87 | 42.42 | 43.32 | 44.47 | 44.97 | 46.85 | 47.81 | 48.69 | **49.10** | 48.40 | 47.60 |
| | Regression Acc | 11.12 | 11.23 | 11.02 | 10.89 | 10.96 | 11.28 | 12.11 | 12.34 | 12.50 | 12.84 | 12.88 | 13.12 | 13.48 | 13.60 | 13.99 | **14.24** | 14.16 | 14.02 | 14.06 |
| 20% | Test Acc | 47.61 | 47.33 | 49.39 | 50.11 | 49.93 | 52.48 | 53.24 | 55.52 | 56.69 | 57.21 | 58.68 | 59.20 | **59.80** | 58.88 | 57.67 | 56.64 | 54.87 | - | - |
| | Regression Acc | 25.62 | 25.58 | 25.78 | 25.76 | 26.48 | 27.00 | 27.36 | 27.59 | 27.67 | 27.85 | 28.17 | 28.29 | **28.51** | 28.44 | 28.23 | 27.93 | 27.30 | - | - |
| 30% | Test Acc | 58.57 | 60.51 | 63.46 | 62.25 | 62.60 | 65.34 | 64.46 | 64.45 | 64.92 | **65.48** | 64.50 | 63.50 | 63.02 | 61.32 | 60.15 | - | - | - | - |
| | Regression Acc | 43.27 | 43.46 | 43.83 | 43.85 | 44.11 | **44.37** | 44.22 | 44.23 | 44.01 | 43.78 | 43.56 | 43.21 | 42.56 | 41.95 | 40.92 | - | - | - | - |
| 40% | Test Acc | 68.44 | 69.92 | 69.53 | 69.17 | **70.81** | 70.14 | 70.39 | 70.47 | 69.06 | 68.04 | 66.66 | 65.40 | 63.86 | - | - | - | - | - | - |
| | Regression Acc | 54.05 | 54.36 | **54.44** | 54.30 | 53.96 | 53.67 | 53.25 | 52.74 | 52.23 | 51.56 | 50.72 | 49.73 | 48.64 | - | - | - | - | - | - |

Table 9: Comparison of window subsets of ImageNet in terms of their 1) test accuracy, measured on models trained with the window subsets (top rows) and 2) accuracy of kernel ridge regression on the training dataset (bottom rows).

| Ratio | Starting point | 0% | 5% | 10% | 15% | 20% | 25% | 30% | 35% | 40% | 45% | 50% | 55% | 60% | 65% | 70% | 75% | 80% | 85% | 90% |
|---|---|---|---|---|---|---|---|---|---|---|---|---|---|---|---|---|---|---|---|---|
| 10% | Test Acc | 46.83 | 45.91 | 46.11 | 46.62 | 44.20 | 46.25 | 45.34 | 47.28 | 48.72 | 47.49 | **48.84** | 47.40 | 46.24 | 46.50 | 46.72 | 45.02 | 43.28 | 39.70 | 32.49 |
| | Regression Acc | 35.05 | 35.45 | 35.71 | 35.84 | 35.74 | 35.72 | 35.89 | 35.99 | 36.03 | 36.06 | 36.05 | 36.04 | 36.19 | 36.23 | **36.24** | 36.17 | 35.90 | 35.01 | 32.43 |
| 20% | Test Acc | 60.55 | 62.32 | 59.67 | **62.83** | 61.27 | 61.89 | 62.19 | 60.92 | 61.28 | 61.17 | 59.42 | 59.50 | 58.73 | 56.86 | 54.70 | 52.21 | 46.59 | - | - |
| | Regression Acc | 44.72 | **44.91** | 44.83 | 44.72 | 44.59 | 44.54 | 44.35 | 44.21 | 44.12 | 43.94 | 43.89 | 43.84 | 43.75 | 43.65 | 43.39 | 42.81 | 41.30 | - | - |
| 30% | Test Acc | 67.16 | **68.22** | 67.62 | 67.98 | 67.82 | 67.91 | 68.19 | 66.84 | 66.78 | 65.54 | 64.67 | 62.62 | 61.73 | 58.72 | 55.50 | - | - | - | - |
| | Regression Acc | **52.19** | 52.14 | 51.93 | 51.70 | 51.52 | 51.26 | 51.04 | 50.83 | 50.65 | 50.46 | 50.30 | 50.14 | 49.86 | 49.39 | 48.36 | - | - | - | - |
| 40% | Test Acc | 70.47 | 70.60 | **71.33** | 71.31 | 70.86 | 70.13 | 69.77 | 69.08 | 68.19 | 67.30 | 66.19 | 63.77 | 61.41 | - | - | - | - | - | - |
| | Regression Acc | **53.48** | 53.41 | 53.23 | 52.98 | 52.74 | 52.54 | 52.27 | 52.05 | 51.87 | 51.67 | 51.42 | 50.99 | 50.14 | - | - | - | - | - | - |

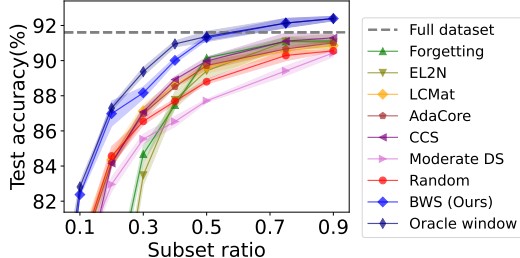

Figure 6: Cross-architecture experiments with EfficientNet-B0, where sample scores are calculated using ResNet18 model. Full results are reported in Table 17.

## F    SLIDING WINDOW EXPERIMENT FOR LABEL-NOISE DATASET

Table 10: Comparison of window subsets of CIFAR-10 dataset with 20% label noise in terms of their 1) test accuracy, measured on models trained with the window subsets (top rows) and 2) accuracy of kernel ridge regression on the training dataset (bottom rows). We also report the noise portion with each window subset. The best window alignment between the two measures gets less accurate, compared to the case without label noise, since our method (regression) tends to choose more easier samples. However, such tendency also makes the choice of window subset mostly composed of clean-label samples.

| Ratio | Starting point | 0% | 5% | 10% | 15% | 20% | 25% | 30% | 35% | 40% | 45% | 50% | 55% | 60% | 65% | 70% | 75% | 80% | 85% | 90% |
|---|---|---|---|---|---|---|---|---|---|---|---|---|---|---|---|---|---|---|---|---|
| | Test Acc | 7.23 | 8.98 | 13.64 | 34.72 | 63.55 | 75.70 | 80.74 | 83.82 | 84.40 | 84.61 | 85.22 | 85.63 | **85.72** | 85.28 | 84.86 | 84.04 | 83.61 | 82.80 | 80.77 |
| 10% | Regression Acc | 51.54 | 55.95 | 58.10 | 59.52 | 60.42 | 61.68 | 63.86 | 64.94 | 66.11 | 67.58 | 68.74 | 69.64 | 69.73 | 70.09 | 70.20 | 70.43 | **70.54** | 70.13 | 69.73 |
| | Noise Portion | 92% | 85% | 69% | 39% | 15% | 8% | 5% | 4% | 4% | 4% | 3% | 3% | 3% | 3% | 3% | 3% | 3% | 3% | 3% |
| | Test Acc | 9.80 | 18.15 | 37.57 | 65.49 | 81.84 | 87.60 | 88.89 | **89.35** | 89.26 | 89.04 | 88.62 | 88.44 | 88.08 | 87.17 | 86.78 | 86.14 | 84.75 | - | - |
| 20% | Regression Acc | 73.28 | 75.17 | 76.06 | 77.03 | 78.38 | 78.66 | 79.05 | 79.63 | 80.01 | 80.47 | 80.67 | 80.67 | 80.66 | 80.63 | **80.70** | 80.60 | 80.35 | - | - |
| | Noise Portion | 80% | 62% | 42% | 24% | 10% | 6% | 5% | 4% | 4% | 3% | 3% | 3% | 3% | 3% | 3% | 3% | 3% | - | - |
| | Test Acc | 21.87 | 39.90 | 64.11 | 81.54 | 89.04 | 90.89 | **91.12** | 90.76 | 90.55 | 89.95 | 89.61 | 89.14 | 88.55 | 88.08 | 87.09 | - | - | - | - |
| 30% | Regression Acc | 80.43 | 81.39 | 82.33 | 83.20 | 83.45 | 83.84 | 84.10 | 84.30 | 84.36 | 84.55 | 84.47 | **84.60** | 84.39 | 84.41 | 84.16 | - | - | - | - |
| | Noise Portion | 59% | 44% | 30% | 17% | 8% | 5% | 4% | 4% | 3% | 3% | 3% | 3% | 3% | 3% | 3% | - | - | - | - |
| | Test Acc | 40.00 | 61.73 | 78.89 | 88.17 | 91.52 | **92.06** | 91.80 | 91.54 | 91.03 | 90.29 | 89.67 | 89.32 | 89.11 | - | - | - | - | - | - |
| 40% | Regression Acc | 86.19 | 86.50 | 86.63 | 86.87 | 86.96 | 87.11 | 87.21 | 87.16 | 87.29 | **87.39** | 87.25 | 87.25 | 87.16 | - | - | - | - | - | - |
| | Noise Portion | 45% | 34% | 23% | 14% | 7% | 5% | 4% | 4% | 3% | 3% | 3% | 3% | 3% | - | - | - | - | - | - |

## G    ADDITIONAL EXPERIMENTS

### G.1    CROSS ARCHITECTURE ROBUSTNESS

In this section, we conduct data pruning experiments on the CIFAR-10 dataset, utilizing the EfficientNet-B0 architecture (Tan & Le, 2019). We create our window subsets from samples ranked by their Forgetting score, calculated using the ResNet18 framework. The best window selection (Alg. 1) and model training processes are performed using the EfficientNet-B0 architecture. We closely follow the implementation details of Tan & Le (2019), by setting the learning rate to $1e-4$ and a weight decay of the same magnitude. Other implementation specifications are the same with the details in Sec. C.2 of CIFAR-10. The results on the EfficientNet-B0 framework are presented in Fig. 6. Our proposed BWS methodology consistently outperforms other benchmark methods, demonstrating its robustness to architecture changes during sample scoring and training.

## G.2 ABLATION STUDY ON DIFFICULTY SCORES

Table 11: Test accuracy of the BWS algorithm at different data selection ratios, depending on the difficulty score. Owing to the high correlation between the difficulty scores, there is a similar sorting order across them and similar window positions are selected, regardless of the specific difficulty score in use. This uniformity in selection leads to consistently strong performance irrespective of the chosen difficulty score.

| Selection methods | Selection ratio | 1% | 5% | 10% | 20% | 30% | 40% | 50% | 75% | 90% |
|---|---|---|---|---|---|---|---|---|---|---|
| BWS with EL2N | Test accuracy | 60.97 | 81.58 | 86.15 | 90.51 | 92.30 | 93.44 | 94.93 | 95.04 | 95.28 |
| | Window index | 85% | 60% | 45% | 25% | 15% | 10% | 0% | 0% | 0% |
| BWS with forgetting | Test accuracy | 63.14 | 81.31 | 88.05 | 91.29 | 93.17 | 94.38 | 94.93 | 95.20 | 95.22 |
| | Window index | 90% | 70% | 35% | 25% | 15% | 5% | 0% | 0% | 0% |
| Oracle window | Test accuracy | 65.73 | 83.03 | 88.05 | 91.69 | 93.35 | 94.38 | 94.93 | 95.20 | 95.22 |
| | Window index | 80% | 50% | 35% | 20% | 10% | 5% | 0% | 0% | 0% |

In the implementation of our BWS algorithm, we employ the Forgetting score (Toneva et al., 2019) as a difficulty score. To test the algorithm's adaptability to alternative difficulty scores, we examine its performance when configured with the EL2N score (Paul et al., 2021). Table 11 compares the results achieved by our BWS algorithm when driven by the EL2N score against those obtained with the Forgetting score. Regardless of whether the algorithm uses the Forgetting score or the EL2N score, both the results achieve competitive performances near that of the oracle window across a wide range of selection ratios. We anticipate that the observed phenomenon arises due to a strong correlation between the difficulty scores. The rank correlation between the EL2N score and the forgetting score used for comparison is notably high as 0.8836. This suggests that samples sorted by the two difficulty scores would likely follow a similar order. As a result, the best windows selected by BWS for the two different score cases exhibit similarity, as shown by Table 11. This consistency shows that the effectiveness of BWS is not limited by the choice of the difficulty score, highlighting its robustness to the sample scores used in sorting.

## G.3 ABLATION STUDY ON WINDOW TYPE: TWO HALF-WIDTH SLIDING WINDOWS

BWS sorts samples in a dataset by their difficulty scores and then selects the optimal window subset from one continuous single-interval regime. Thus, the window selection chooses the samples of similar difficulty level. Our ablation study in Table 3 compares different types of subsets (in addition to the hard-only or easy-only windows), such as "Hard-easy", which combines the easy and hard samples with equal portions, or "25-75%", which includes random samples from a moderate regime after pruning a fixed portion of too easy or hard samples. Compared to these non-contiguous subset selection methods, our window selection consistently achieves better performance over all the selection ratios (10 to 40%) by cleverly choosing the window starting point.

To further examine any possible benefit from non-contiguous subset selection, we conduct an additional experiment on the CIFAR-10 dataset by finding the optimal two half-width windows while varying their starting points. In detail, we sort the samples from CIFAR-10 in descending order based on Forgetting score (Toneva et al., 2019) similar to Sec. 4 and for a subset of size $w\%$, we search over all combinations of two half-width windows, denoted by $[x_1, x_1 + w/2] \cup [x_2, x_2 + w/2]$ while varying their starting points $(x_1, x_2)$ in $x_1 \in [0, 100 - w]$ and $x_2 \in [x_1 + w/2, 100 - w/2]$ with a step size of 5%. We train ResNet18 on each subset and evaluate the corresponding test accuracies. The full results are presented in Fig. 7, and in Table 12 we report the top five results (the compositions of half-width windows and their test accuracies) for subset ratios ranging from 10 to 40%. We highlight the cases where the two half-width windows are contiguous to each other with bold letters.

We can observe that for every considered subset ratio, the top-five best performing cases include contiguous windows (or windows near to each other with the gap of only 5%), even though we allowed flexibility in choosing the two half-width windows far away from each other. This result further supports our use of window selection, which only considers subsets from a continuous interval of samples based on difficulty scores, in choosing near-optimal subset in an efficient manner across a broad range of selection ratios.

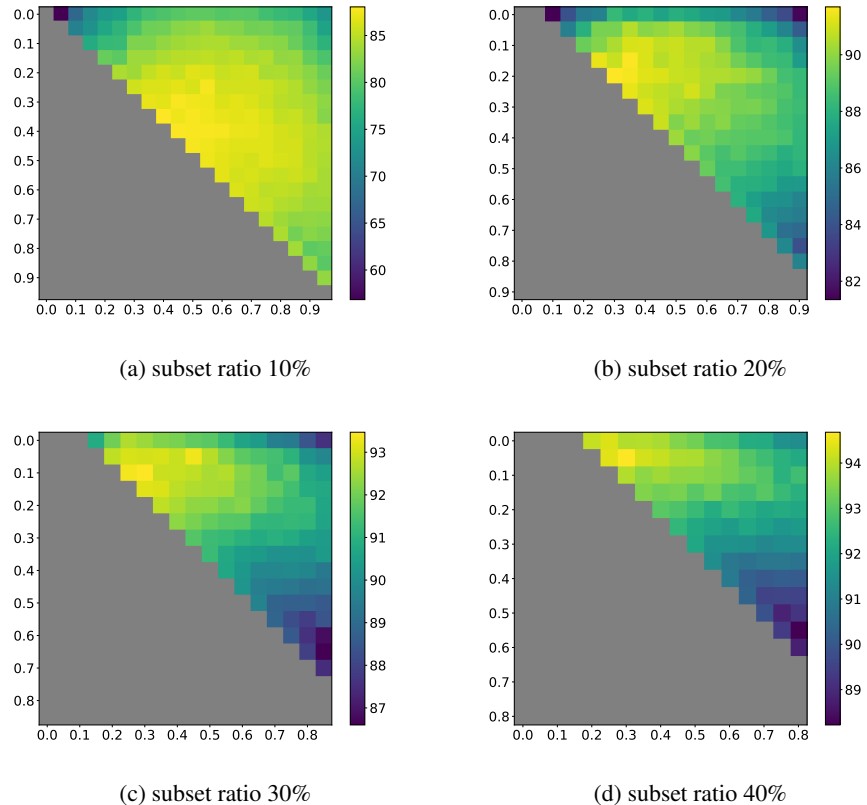

(a) subset ratio 10%

(b) subset ratio 20%

(c) subset ratio 30%

(d) subset ratio 40%

Figure 7: Test accuracy of the models trained with two half-width windows of varying starting points. The numbers in axes indicate the starting points of each interval, and the color indicates the test accuracy for each composition of half-width windows. We can observe that the contiguous windows (or windows near to each other) achieve one of the best performances.

Table 12: Top-five test accuracies and their corresponding half-width window compositions on CIFAR-10 dataset. We highlight the cases where the two half-width windows are contiguous to each other with bold letters.

| Ratio | Ranking | 1st | 2nd | 3rd | 4th | 5th |
|---|---|---|---|---|---|---|
| 10% | Half-width windows | **35-40%, 40-45%** | **40-45%, 45-50%** | 30-35%, 45-50% | 40-45%, 50-55% | 40-45%, 55-60% |
| | Test Acc | **88.05** | **87.82** | 87.80 | 87.74 | 87.71 |
| 20% | Half-width windows | **20-30%, 30-40%** | 15-25%, 35-45% | 20-30%, 35-45% | **15-25%, 25-35%** | **25-35%, 35-45%** |
| | Test Acc | **91.69** | 91.61 | 91.56 | **91.34** | **91.29** |
| 30% | Half-width windows | 10-25%, 30-45% | **10-25%, 25-40%** | 5-20%, 45-60% | **15-30%, 30-45%** | 15-30%, 35-50% |
| | Test Acc | 93.47 | **93.35** | 93.31 | **93.17** | 93.11 |
| 40% | Half-width windows | 5-25%, 30-50% | **5-25%, 25-45%** | 0-20%, 25-45% | 5-25%, 35-55% | 5-25%, 40-60% |
| | Test Acc | 94.68 | **94.38** | 94.35 | 94.23 | 94.11 |

## H  FULL RESULTS

In Table 13, 14, 15, and 16, the oracle window achieves the highest performance among the considered methodologies for almost every selection ratio and dataset, since it finds the best window by directly measuring and comparing the test accuracy of models trained by each window using the test dataset. Since the oracle window cannot be implemented in practice, it is fair to compare the performances among the methods except the oracle window. Thus, we highlight the highest and the second-highest values among the rest of the methodologies except the oracle window.

Table 13: Test accuracy of CIFAR-10 dataset. We highlight the highest values in bold and the second-highest values in underscore.

| Selection ratio | 1% | 5% | 10% | 20% | 30% | 40% | 50% | 75% | 90% | 100% |
|---|---|---|---|---|---|---|---|---|---|---|
| Forgetting | 30.56±1.28 | 45.86±3.44 | 58.88±1.90 | 81.29±0.62 | 90.88±0.25 | 94.23±0.03 | 94.92±0.07 | 95.17±0.07 | 95.11±0.05 | |
| EL2N | 16.10±0.65 | 28.71±0.55 | 43.53±1.62 | 74.91±0.67 | 90.33±0.07 | 93.81±0.01 | 94.86±0.16 | 95.38±0.03 | 95.39±0.11 | |
| LCMat | 51.24±1.05 | 78.15±0.59 | 84.06±0.15 | 89.16±0.37 | 91.82±0.22 | 93.11±0.11 | 93.74±0.15 | 94.86±0.11 | 95.26±0.16 | |
| AdaCore | 50.94±2.23 | 77.94±0.52 | 84.04±0.09 | 89.03±0.35 | 91.74±0.12 | 93.27±0.23 | 93.76±0.12 | 94.92±0.05 | 95.25±0.23 | |
| CCS | 33.58±0.61 | 71.15±1.43 | 81.56±0.23 | 89.28±0.34 | 92.50±0.16 | 93.98±0.12 | 94.78±0.12 | 95.24±0.16 | 95.32±0.15 | 95.46±0.02 |
| Moderate DS | 54.76±2.22 | 77.51±0.38 | 83.61±0.26 | 87.86±0.03 | 89.90±0.07 | 91.33±0.03 | 92.04±0.16 | 93.97±0.01 | 94.75±0.14 | |
| Random | 49.59±5.75 | 77.35±0.11 | 84.14±0.42 | 89.15±0.13 | 91.10±0.33 | 92.41±0.28 | 93.29±0.15 | 94.60±0.20 | 95.01±0.09 | |
| BWS (Ours) | 63.14±1.06 | 81.31±0.38 | 88.05±0.04 | 91.29±0.06 | 93.17±0.02 | 94.38±0.12 | 94.93±0.07 | 95.20±0.10 | 95.22±0.19 | |
| Oracle window | 65.73±1.09 | 83.03±0.14 | 88.05±0.04 | 91.69±0.18 | 93.35±0.16 | 94.38±0.12 | 94.93±0.07 | 95.20±0.10 | 95.22±0.19 | |

Table 14: Test accuracy of CIFAR-100 dataset. We highlight the highest values in bold and the second-highest values in underscore.

| Selection ratio | 1% | 5% | 10% | 20% | 30% | 40% | 50% | 75% | 90% | 100% |
|---|---|---|---|---|---|---|---|---|---|---|
| Forgetting | 11.71±0.10 | 23.19±0.93 | 34.32±1.71 | 48.83±0.26 | 59.11±2.10 | 66.18±1.17 | 71.67±1.19 | 77.43±0.18 | 78.33±0.72 | |
| EL2N | 5.32±0.10 | 8.71±0.07 | 14.30±0.85 | 27.88±0.66 | 42.87±1.15 | 57.68±1.76 | 67.88±1.68 | 76.65±0.23 | 78.02±0.44 | |
| LCMat | 16.16±0.33 | 35.21±0.83 | 46.80±0.82 | 57.25±0.04 | 63.28±1.46 | 67.82±0.85 | 71.74±0.64 | 76.66±0.52 | 78.01±0.56 | |
| AdaCore | 10.36±0.27 | 28.90±0.27 | 42.48±1.80 | 56.06±0.08 | 63.79±0.97 | 69.50±0.44 | 71.93±0.24 | 75.36±0.19 | 78.30±0.34 | |
| CCS | 12.20±0.58 | 32.73±0.33 | 44.63±2.13 | 56.15±0.85 | 63.16±1.24 | 66.77±0.70 | 69.95±0.30 | 75.67±0.35 | 77.57±0.39 | 78.96±0.26 |
| Moderate DS | 14.01±2.35 | 32.55±2.92 | 46.59±2.54 | 58.01±1.49 | 63.97±0.37 | 68.51±1.12 | 70.80±1.10 | 75.74±0.86 | 77.84±0.61 | |
| Random | 11.25±0.69 | 30.97±1.43 | 41.76±1.43 | 56.33±0.25 | 64.09±0.24 | 68.10±0.62 | 70.57±0.59 | 76.16±0.15 | 77.65±0.57 | |
| BWS (Ours) | 17.20±0.12 | 39.13±0.37 | 48.69±0.41 | 59.80±0.49 | 65.34±1.04 | 69.53±1.11 | 72.27±1.06 | 78.20±0.40 | 77.96±0.17 | |
| Oracle window | 18.92±0.54 | 39.33±1.64 | 49.10±0.30 | 59.80±0.49 | 65.48±0.53 | 70.81±0.97 | 74.11±0.33 | 78.20±0.40 | 78.08±0.39 | |

Table 15: Test accuracy of ImageNet dataset. We highlight the highest values in bold and the second-highest values in underscore.

| Selection ratio | 1% | 5% | 10% | 20% | 30% | 40% | 50% | 75% | 90% | 100% |
|---|---|---|---|---|---|---|---|---|---|---|
| Forgetting | 4.78±0.10 | 28.18±0.46 | 45.84±0.67 | 60.75±0.60 | 67.48±0.11 | 70.26±0.48 | 72.73±0.09 | 74.63±0.13 | 75.53±0.06 | |
| EL2N | 2.10±0.08 | 9.80±0.03 | 20.42±0.47 | 41.14±0.04 | 54.42±0.39 | 63.19±0.29 | 68.19±0.13 | 73.91±0.36 | 74.79±0.27 | |
| LCMat | 6.01±0.31 | 32.26±0.84 | 46.08±0.64 | 59.02±0.36 | 65.28±0.21 | 68.50±0.56 | 70.30±0.46 | 74.13±0.12 | 74.81±0.02 | |
| AdaCore | 6.01±0.44 | 31.52±0.58 | 46.98±0.80 | 59.26±1.58 | 65.18±0.05 | 68.28±0.05 | 70.72±0.04 | 73.53±0.13 | 74.69±0.00 | |
| CCS | 5.04±0.40 | 31.83±0.62 | 46.64±1.08 | 58.77±0.80 | 64.85±0.12 | 67.82±0.24 | 69.89±0.24 | 73.57±0.12 | 74.59±0.03 | 75.85±0.07 |
| Moderate DS | 5.97±0.60 | 32.47±0.21 | 47.83±0.11 | 58.86±0.14 | 64.71±0.41 | 67.47±0.03 | 69.73±0.08 | 73.16±0.25 | 74.67±0.07 | |
| Random | 6.14±0.01 | 33.17±0.11 | 45.87±0.07 | 59.19±0.04 | 65.94±0.38 | 68.23±0.00 | 70.14±0.31 | 73.74±0.14 | 74.83±0.08 | |
| BWS (Ours) | 7.02±0.10 | 33.31±0.70 | 46.72±0.21 | 62.32±0.47 | 67.16±0.12 | 70.47±0.17 | 72.68±0.14 | 74.73±0.04 | 75.25±0.34 | |
| Oracle window | 7.97±0.29 | 33.58±0.11 | 48.84±0.34 | 62.83±0.08 | 68.22±0.69 | 71.33±0.29 | 72.74±0.36 | 74.73±0.04 | 75.25±0.34 | |

Table 16: Test accuracy of CIFAR-10 dataset by training a simple CNN architecture. We highlight the highest values in bold and the second-highest values in underscore.

| Selection ratio | 1% | 5% | 10% | 20% | 30% | 40% | 50% | 75% | 90% | 100% |
|---|---|---|---|---|---|---|---|---|---|---|
| Forgetting | 35.33±0.45 | 43.33±0.64 | 51.90±0.59 | 67.87±0.28 | 76.76±0.43 | 81.60±0.59 | 84.34±0.73 | 87.16±0.60 | 87.71±0.15 | |
| EL2N | 18.06±0.13 | 31.04±0.36 | 42.59±0.53 | 63.19±0.25 | 75.15±0.47 | 81.13±0.85 | 84.39±1.02 | 87.00±0.50 | 87.23±0.35 | |
| LCMat | 49.72±0.27 | 66.91±0.32 | 73.14±0.80 | 79.12±0.74 | 82.10±0.76 | 84.32±0.57 | 85.58±0.35 | 87.20±0.50 | 87.68±0.45 | |
| AdaCore | 49.72±0.27 | 66.91±0.32 | 73.14±0.80 | 79.12±0.74 | 82.10±0.76 | 84.32±0.57 | 85.58±0.35 | 87.20±0.50 | 87.68±0.45 | |
| CCS | 39.42±0.46 | 58.84±0.27 | 68.37±0.48 | 76.90±0.48 | 81.41±0.41 | 84.08±0.45 | 85.44±0.22 | 87.17±0.18 | 87.87±0.33 | 88.10±0.23 |
| Moderate DS | 50.26±0.26 | 67.99±0.24 | 72.66±0.53 | 78.07±0.60 | 80.90±0.81 | 82.47±0.59 | 83.34±0.60 | 85.72±0.41 | 86.84±0.57 | |
| Random | 48.99±1.13 | 66.13±0.23 | 73.87±0.70 | 79.71±0.68 | 82.36±0.60 | 83.91±0.45 | 84.91±0.86 | 87.12±0.36 | 87.39±0.33 | |
| BWS (Ours) | 57.49±0.69 | 72.15±0.33 | 78.34±0.19 | 81.56±0.41 | 83.74±0.33 | 84.91±0.62 | 85.76±0.05 | 87.45±0.13 | 87.71±0.25 | |
| Oracle window | 57.49±0.69 | 73.43±0.19 | 78.34±0.19 | 81.99±0.41 | 83.98±0.39 | 84.92±0.13 | 86.17±0.36 | 87.45±0.13 | 87.71±0.25 | |

Table 17: Test accuracy of CIFAR-10 dataset by training EfficientNet-B0 architecture. We highlight the highest values in bold and the second-highest values in underscore.

| Selection ratio | 1% | 5% | 10% | 20% | 30% | 40% | 50% | 75% | 90% | 100% |
|---|---|---|---|---|---|---|---|---|---|---|
| Forgetting | 28.34±0.82 | 37.65±1.11 | 56.51±2.43 | 76.22±0.48 | 84.68±0.48 | 87.47±0.18 | 90.15±0.06 | 91.09±0.22 | 91.12±0.08 | |
| EL2N | 15.63±0.67 | 26.93±1.39 | 44.75±2.73 | 72.31±0.39 | 83.48±0.57 | 87.77±0.49 | 89.42±0.50 | 91.09±0.26 | 91.04±0.44 | |
| LCMat | 45.40±3.60 | 71.91±0.83 | 79.07±0.77 | 84.35±0.42 | 87.13±0.37 | 88.66±0.01 | 89.64±0.35 | 90.56±0.27 | 90.85±0.20 | |
| AdaCore | 47.97±2.37 | 72.62±0.86 | 78.80±0.77 | 84.21±0.23 | 86.98±0.36 | 88.52±0.03 | 89.72±0.23 | 90.67±0.13 | 91.01±0.11 | |
| CCS | 30.84±1.04 | 62.51±2.86 | 76.36±0.89 | 84.15±0.20 | 87.08±0.23 | 88.92±0.18 | 89.98±0.21 | 91.11±0.46 | 91.30±0.20 | 91.61±0.12 |
| Moderate DS | 45.98±3.67 | 73.09±0.45 | 78.29±0.11 | 82.96±0.40 | 85.54±0.28 | 86.53±0.29 | 87.72±0.09 | 89.42±0.28 | 90.44±0.23 | |
| Random | 47.37±1.28 | 68.74±2.08 | 79.03±0.28 | 84.57±0.54 | 86.55±0.21 | 87.69±0.14 | 88.80±0.14 | 90.29±0.30 | 90.56±0.03 | |
| BWS (Ours) | 56.40±0.79 | 75.06±0.48 | 82.38±0.09 | 86.98±0.72 | 88.17±0.33 | 90.01±0.17 | 91.30±0.15 | 92.14±0.27 | 92.39±0.13 | |
| Oracle window | 56.40±0.79 | 76.43±1.68 | 82.81±0.27 | 87.30±0.15 | 89.37±0.25 | 90.95±0.17 | 91.37±0.30 | 92.14±0.27 | 92.39±0.13 | |

Table 18: Test accuracy of CIFAR-10 dataset by fine-tuning ViT pretrained using ImageNet. We highlight the highest values in bold and the second-highest values in underscore.

| Selection ratio | 1% | 5% | 10% | 20% | 100% |
|---|---|---|---|---|---|
| Forgetting | 93.75±1.10 | 98.00±0.14 | 98.45±0.03 | 98.67±0.06 | |
| EL2N | 84.63±1.77 | 96.59±0.14 | 97.83±0.14 | 98.42±0.10 | |
| LCMat | 93.32±0.42 | 97.07±0.32 | 97.83±0.10 | 98.28±0.04 | |
| AdaCore | 94.45±0.54 | 97.05±0.12 | 97.88±0.11 | 98.28±0.09 | |
| CCS | 94.83±0.31 | 97.84±0.08 | 98.23±0.08 | 98.52±0.03 | 98.76±0.02 |
| Moderate DS | 94.73±0.58 | 97.22±0.06 | 97.69±0.08 | 98.04±0.04 | |
| Random | 94.87±0.67 | 97.47±0.06 | 97.89±0.09 | 98.14±0.06 | |
| BWS (Ours) | **95.47±0.37** | **98.04±0.32** | **98.45±0.01** | **98.70±0.05** | |
| Oracle window | 97.22±0.30 | 98.18±0.21 | 98.45±0.01 | 98.70±0.05 | |

Table 19: Test accuracy of 20% noise CIFAR-10 dataset. We highlight the highest values in bold and the second-highest values in underscore.

| Selection ratio | 10% | 20% | 30% | 40% | 100% |
|---|---|---|---|---|---|
| Forgetting | 53.64±0.73 | 64.92±0.99 | 70.97±0.11 | 74.25±0.26 | |
| EL2N | 6.53±0.10 | 9.73±0.33 | 21.65±0.24 | 39.42±0.44 | |
| LCMat | 63.70±2.10 | 75.99±0.66 | 80.90±0.69 | 82.61±0.43 | |
| AdaCore | 10.71±0.46 | 11.09±0.26 | 39.61±0.13 | 60.37±0.78 | |
| CCS | 62.43±2.03 | 73.98±1.15 | 79.28±0.04 | 81.70±0.14 | 86.45±0.47 |
| Moderate DS | 83.55±0.32 | **88.90±0.08** | **90.97±0.18** | **92.27±0.08** | |
| Random | 72.46±0.71 | 78.99±0.49 | 82.16±0.28 | 83.73±0.54 | |
| BWS (Ours) | **83.61±0.34** | 86.78±0.03 | 89.14±0.24 | 90.29±0.10 | |
| Oracle window | 85.72±0.15 | 89.35±0.09 | 91.12±0.16 | 92.06±0.05 | |

