# OpenReview forum: "BWS: Best Window Selection Based on Sample Scores for Data Pruning across Broad Ranges"
_ICLR.cc/2024/Conference — Submitted to ICLR 2024_

### Official Review · Reviewer_LWro · 2023-10-30

**Soundness:** 3 good
**Presentation:** 3 good
**Contribution:** 3 good
**Rating:** 6
**Confidence:** 4

**Summary:**

This paper proposes a method that aims to find informative subset of the original datasets which can be used to train the neural networks with small performance drop compared with model trained with whole dataset. The point of this paper is to propose a method that can do both universal and efficient selection of subset based on the difficulty score. To adaptively select the best subset, the authors propose a method based on kernel ridge regression. The proposed method can be used to select subset for both training from scratch and fine-tuning. Extensive experiments are conducted to verify the efficacy of the proposed method.

**Strengths:**

This paper gives a deep understanding of which kind of data can be useful for different size of subset and use kernel regression to analyze this problem theoretically.
The situations  for hard sample and easy sample  to have benign effect is reasonable.
The usage of kernel ridge regression for subset selection is interesting.  The details of each parts of the proposed method are illustrated clearly.
Extensive experiments validate the efficacy of the proposed method for training from scratch.
The method can also be effective when used to select subset for fine-tuning.
Ablation studies also validate the robustness of the proposed method.

**Weaknesses:**

For the experiments on CIFAR-10 with noise, the proposed method is outperformed by Moderate DS for 3 ratios. Could the authors illustrate the noisy rate of the selected subset to check whether the proposed method is prone to choose noisy data under this setting?

The experiments on CIFAR-10 fine-tuning on VIT shows that CCS is consistently better than the proposed method, could the author give concrete analysis of this phenomenon?

**Questions:**

Please refer to weakness.

---

> ### Author Response · Authors · 2023-11-16
> **Response to Reviewer LWro**
>
> We sincerely thank the reviewer for the constructive feedback. Hope this response answers the reviewer’s questions.
>
> >**1. For the experiments on CIFAR-10 with noise, the proposed method is outperformed by Moderate DS for 3 ratios. Could the authors illustrate the noisy rate of the selected subset to check whether the proposed method is prone to choose noisy data under this setting?**
>
> Below we compare the noisy rate of the selected subset (the fraction of label-noise data in the subset) between our method (BWS) and Moderate DS for the label-noise experiment reported in Fig. 4(c\), considering 20\% label-noise in CIFAR-10 dataset. We can see that the noise rate is comparable or relatively lower for Moderate DS across the subset ratios (10-40\%), which could be a potential reason for Moderate DS’s outperformance for some cases.
>
> | Selection methods | Subset ratio  | 10%   | 20%   | 30%   | 40%   |
> |:-----------------:|:-------------:|:-----:|:-----:|:-----:|:-----:|
> | BWS               | Test accuracy | 83.61 | 86.78 | 89.14 | 90.29 |
> |                   | Noise ratio   | 3.1%  | 2.7%  | 2.8%  | 3.7%  |
> | Moderate DS       | Test accuracy | 83.55 | 88.9  | 90.97 | 92.27 |
> |                   | Noise ratio   | 2.4%  | 2.4%  | 2.6%  | 2.6%  |
>
> To further compare these two methods, we additionally conducted a label-noise experiment by increasing the noise rate to 40\%. We remind that Moderate-DS selects samples closest to the median of the features of each class, which can be interpreted as selecting moderate difficulty samples first. When the noise rate is as high as 40\%, the top 40\% difficult samples include many label-noise samples. Thus, Moderate-DS starts to include these noisy samples when the subset ratio becomes higher than a certain portion. On the other hand, since our method solves the proxy task to choose the best window subset of the highest accuracy while varying the starting point of the window, our method is still robust even for high noise ratio and relatively large subset portion. The table below compares the two methods in this scenario, and we can see that the performance of Moderate-DS degrades significantly at the subset ratio of 40\%, while our method maintains its performance.
>
> | Selection methods | Subset ratio  | 10%   | 20%   | 30%   | 40%   |
> |:-----------------:|:-------------:|:-----:|:-----:|:-----:|:-----:|
> | BWS               | Test accuracy | 83.89 | 88.02 | 90.07 | 90.58 |
> |                   | Noise ratio   | 7.5%  | 7.1%  | 7.9%  | 8.3%  |
> | Moderate DS       | Test accuracy | 83.95 | 89.16 | 90.1  | 85.63 |
> |                   | Noise ratio   | 6.5%  | 6.7%  | 8.8%  | 19.9% |
>
>
> >**2. The experiments on CIFAR-10 fine-tuning on VIT shows that CCS is consistently better than the proposed method, could the author give concrete analysis of this phenomenon?**
>
> We apologize for the confusion originated from our previous report for CCS on VIT. There was some unintentional inconsistency in our experimental setup and we found that the pretrained ViT parameters were different (by using different versions of timm library) only for CCS in our previous experiment on VIT, which caused significantly better outcomes for CCS compared to other methodologies for some subset ratios. We have updated the paper with the revised results by matching the experimental conditions for CCS with other algorithms, and found that our method (BWS) outperforms CCS on the fine-tuning of VIT as summarized in the table below.
>
> | subset ratio      | 1%             | 5%             | 10%            | 20%            |
> |:-----------------:|:--------------:|:--------------:|:--------------:|:--------------:|
> | Updated CCS       | 94.83   | 97.84  | 98.23 | 98.52 |
> | BWS (Ours)        | 95.47   | 98.04   | 98.45   | 98.70  |

---

> > ### Comment · Reviewer_LWro · 2023-11-23
> >
> > I appreciate the response, I will keep my score.

---

### Official Review · Reviewer_BL3J · 2023-10-31

**Soundness:** 2 fair
**Presentation:** 3 good
**Contribution:** 2 fair
**Rating:** 5
**Confidence:** 4

**Summary:**

The paper proposes a novel and universal coreset selection method called "Best Window Selection (BWS)" to strike a balance between sample diversity and model performance across a broad range of selection ratios. BWS first sorts all training examples w.r.t. the difficulty score and then prunes a specific number of the most difficult examples and easiest examples. By comparing BWS with other SOTA baselines, the evaluation results show that BWS outperforms other coreset selection methods.

**Strengths:**

1. This paper proposes a novel coreset selection method, BWS. Compared to previous work, BWS selects the best window more efficiently with kernel ridge regression, which is faster than training a model from scratch.

2. The evaluation results show that BWS achieves better or comparable results to other SOTA methods.

3. The overall writing is good and easy to follow.

**Weaknesses:**

1. Using kernel ridge regression to decide the best window is not quite intuitive. What is the motivation to use kernel ridge regression rather than training a small network to decide the best window?

2. The baseline evaluation results are inconsistent with data reported in the baseline method. For example, moderate are reported to have better performance than random on CIFAR10. CCS seems to have a better performance at 10% subset ratio than the numbers reported in the paper. It may be good to explain why the difference exists.

**Questions:**

I don’t fully understand why the performance of $w_s$ can represent the performance of models trained on the same subset. Could the authors further explain the connection between kernel regression and deep learning model training? What I currently feel is that it is more like an empirical transferability stuff studied in [1]: it is possible to use a small model to select coresets that transfer well to larger models.

[1] Coleman, C., et al. "Selection via Proxy: Efficient Data Selection for Deep Learning." International Conference on Learning Representations (ICLR). 2020.

---

> ### Author Response · Authors · 2023-11-16
> **Response to Reviewer BL3J (1/3)**
>
> We sincerely thank the reviewer for the feedback. Hope this response answers the reviewer’s questions.
>
> >**1. Using kernel ridge regression to decide the best window is not quite intuitive. What is the motivation to use kernel ridge regression rather than training a small network to decide the best window?**
>
>
> We appreciate the reviewer for addressing this important question. Our use of the kernel ridge regression as a proxy for training neural networks can be partly explained by the recent progress in theoretical understanding of training neural networks using kernel methods. In particular, some recent works [a,b,c,d] have shown that training and generalization of neural networks can be approximated by two associated kernel matrices: the Conjugate Kernel (CK) and Neural Tangent Kernel (NTK). The Conjugate Kernel is defined by the gram matrix of the derived features produced by the final hidden layer of the network, while NTK is the gram matrix of the Jacobian of in-sample predictions with respect to the network weights. These two kernels also have fundamental relations in terms of their eigenvalue distributions as analyzed in [e]. Our proxy task is motivated by observation that the kernel regression with these model-related kernels can provide a good approximation to the original model (under some assumptions such as enough width, random initialization, and small enough learning rate, etc.).
>
> In particular, we use the Conjugate Kernel (CK) in our kernel ridge regression (Equation (2)), by defining the kernel matrix as ${\mathbf{X}\_\mathbf{S}}^\top {\mathbf{X}\_\mathbf{S}}$ where $\mathbf{X}\_\mathbf{S}=[\mathbf{f}\_1,\dots,\mathbf{f}\_m]$ is composed of features produced by the exact target network of our consideration (ResNet18 for CIFAR-10 and ResNet50 for CIFAR-100/ImageNet). By considering the features from the target network, we can obtain the (approximate) network predictions that are linear in these derived features. Of course, this kernel approximation of the neural network models, which assumes a fixed feature extractor, does not exactly match our situation where the selected subset not only affects the linear classifier but also the feature extractor itself during the training. However, we’d like to emphasize that this is a proxy that can reflect the network architecture of our interest in a computationally-efficient manner. Also, our analysis in Table 2 shows that this proxy finds the best window subset that aligns well with the result from the actual training of the full model.
>
> [a] Radford M Neal. Bayesian learning for neural networks, 1995.
>
> [b] Jaehoon Lee, Yasaman Bahri, Roman Novak, Samuel S Schoenholz, Jeffrey Pennington, and Jascha Sohl-Dickstein. Deep neural networks as Gaussian processes, ICLR 2018.
>
> [c] Sanjeev Arora, Simon S. Du, Wei Hu, Zhiyuan Li, Ruslan Salakhutdinov, and Ruosong Wang. On exact computation with an infinitely wide neural net, NeurIPS 2019.
>
> [d] Arthur Jacot, Franck Gabriel, and Clement Hongler. Neural Tangent Kernel: Convergence and Generalization in Neural Networks, NeurIPS 2018.
>
> [e] Zhou Fan and Zhichao Wang. Spectra of the Conjugate Kernel and Neural Tangent Kernel for Linear-Width Neural Networks, NeruIPS 2020.

---

> ### Author Response · Authors · 2023-11-16
> **Response to Reviewer BL3J (2/3)**
>
> >**2. I don’t fully understand why the performance of $w\_s$ can represent the performance of models trained on the same subset. Could the authors further explain the connection between kernel regression and deep learning model training? What I currently feel is that it is more like an empirical transferability stuff studied in [1]: it is possible to use a small model to select coresets that transfer well to larger models.
> [1] Coleman, C., et al. "Selection via Proxy: Efficient Data Selection for Deep Learning." International Conference on Learning Representations (ICLR). 2020.**
>
> As the reviewer suggested, we can also consider an alternative proxy from (SVP, Section via Proxy) by evaluating the performance of each subset using a smaller network. To evaluate the efficiency of this approach, we conducted an additional experiment of finding the best window subset using a small ConvNet model on the CIFAR-10 dataset. The selected subset is then evaluated on ResNet18. The table below summarizes this result (SVP) compared to our original proxy (KRR, Kernel Ridge Regression) and the oracle window using the target model. The table below summarizes the result.
>
> We can observe that SVP tends to select easier samples (windows with larger index) compared to the oracle window over the selection ratios 10\% to 90\%, which results in performance loss especially in high selection ratio regimes. Our proxy, on the other hand, exactly matches the oracle window performance at selection ratios of 10\% to 90\%. We conjecture that the tendency that SVP selects an easier subset is attributed to the limited capability of the simple network used in the proxy task.  Moreover, we’d like to highlight that solving our proxy task takes only about 1/15-1/250 (varying depending on the subset size) of the computational time compared to SVP, which requires training of the small network (ConvNet) for all considered subsets. We added this result in Appendix B of our revised paper.
>
>
>
> | Selection methods | Selection ratio | 1%    | 5%    | 10%   | 20%   | 30%   | 40%   | 50%   | 75%   | 90%   |
> |:-----------------:|:---------------:|:-----:|:-----:|:-----:|:-----:|:-----:|:-----:|:-----:|:-----:|:-----:|
> | KRR (Ours)        | Test accuracy   | 65.14 | 81.31 | 88.05 | 91.29 | 93.17 | 94.38 | 94.93 | 95.2  | 95.22 |
> |                   | Window index    | 90%   | 70%   | 35%   | 25%   | 15%   | 5%    | 0%    | 0%    | 0%    |
> | SVP               | Test accuracy   | 65.73 | 81.78 | 86.4  | 90.18 | 91.63 | 92.5  | 93.07 | 94.96 | 95.22 |
> |                   | Window index    | 80%   | 60%   | 55%   | 40%   | 30%   | 25%   | 20%   | 5%    | 0%    |
> | Oracle window     | Test accuracy   | 65.73 | 83.03 | 88.05 | 91.69 | 93.35 | 94.38 | 94.93 | 95.2  | 95.22 |
> |                   | Window index    | 80%   | 50%   | 35%   | 20%   | 10%   | 5%    | 0%    | 0%    | 0%    |

---

> ### Author Response · Authors · 2023-11-16
> **Response to Reviewer BL3J (3/3)**
>
> >**3. The baseline evaluation results are inconsistent with data reported in the baseline method. For example, moderate are reported to have better performance than random on CIFAR10. CCS seems to have a better performance at 10% subset ratio than the numbers reported in the paper. It may be good to explain why the difference exists.**
>
> The differences between our CCS [f] numbers and those reported in the original paper stem from variations in the hyperparameter, $\beta$, employed within the algorithm. CCS prunes a $\beta$\% of hard examples, with $\beta$ being a hyperparameter, and then selects samples with a uniform difficulty score distribution. Considering $\beta$ as a one-sided threshold for the window subset, tuning this parameter can be viewed as an oracle window search within our algorithm. Thus, for a fair comparison we set beta to 0 for all the pruning ratios, and this makes our evaluation of CCS the same as the 'stratified only' algorithm in Table 2 of the CCS paper. When we set beta to 0, it operates optimally when the subset ratio is high but sub-optimally when the ratio is low. This observation also aligns with the findings in the CCS paper Fig 6(b). Below are the comparisons between the reported numbers for CCS in CIFAR-10 dataset.
>
> | Subset ratio                              | 10%   | 20%   | 30%   | 50%   |
> |:-----------------------------------------:|:-----:|:-----:|:-----:|:-----:|
> | CCS in original paper                     | 85.7  | 90.93 | 92.97 | 95.04 |
> | Stratified only(beta 0) in original paper | 59.23 | 81.82 | 90.78 | 95.13 |
> | CCS in Ours                               | 81.56 | 89.28 | 92.5  | 94.78 |
>
> For the moderate coreset [g], the results for CIFAR10 were not provided in the original paper, so we could not directly compare the numbers. In the case of CIFAR100, we reported better results as summarized in the table below. This discrepancy arises from differences in experimental settings; we preserve the number of iterations per epoch regardless of the subset ratio, whereas the original paper varies the iterations according to the subset ratio.
>
>
> | Subset ratio                  | 20%   | 30%   | 40%   |
> |:-----------------------------:|:-----:|:-----:|:-----:|
> | Moderate DS in original paper | 51.83 | 57.79 | 64.92 |
> | Moderate DS in Ours           | 58.01 | 63.97 | 68.51 |
>
> [f] Coverage-centric Coreset Selection for High Pruning Rates. H Zheng et al.
>
> [g] Moderate Coreset: A Universal Method of Data Selection for Real-world Data-efficient Deep Learning. X Xia et al.

---

> ### Comment · Reviewer_BL3J · 2023-11-22
> **Thanks for the detailed response**
>
> Thank the authors for the response, and I am sorry for my delayed response. The author’s response addresses some of my questions, but some of my concerns still remain:
>
> Although the response explains that SVP always selects easier samples, which is different from BWS, I still feel that BWS shares the same insights as SVP: BWS uses light machine models as a proxy to select the best window for coreset selection. This is also based on the hypothesis that the coreset selected by the proxy models transfers well to the target model, which can impact the novelty of the work.
>
> The additional evaluation provided by the authors shows that BWS actually underperforms SOTA methods (it seems that the number reported is based on a different setting from the original paper). My understanding is that CCS needs more time to choose the Oracle window, but BTW can choose the window in a more efficient way with a trade-off on the drop of accuracy? If this is true, it can hurt the contribution of the work, and the paper should have a more explicit discussion of this trade-off.

---

> ### Author Response · Authors · 2023-11-22
>
> Thank the reviewer for the comments. First, we’d like to point out that the main contributions of our paper are in two parts: 1) reducing the search space of coresets from $n\choose k$ to a constant number by considering the window subsets of a fixed step size and 2) providing a proxy task that can reflect the network architecture of our interest in a computationally-efficient manner. Furthermore, our experimental results demonstrate the superior performance of our approach compared to other baselines across a broad range of selection ratios over datasets, including CIFAR-10/100 and ImageNet, and the scenarios involving training from random initialization or fine-tuning of pre-trained models.
>
> We’d like to further emphasize two crucial differences between SVP and BWS. First, unlike SVP that uses a small network and therefore can not fully reflect the information of the target model, BWS incorporates this information by utilizing features from the model of interest. Second, while training an (even simple) neural network for every different subset is computationally costly, BWS offers a highly efficient way in choosing the best window as solving a regression problem requires only a few seconds. We’d like to point out that our research is the first work theoretically demonstrating that the difficulty level of the optimal subset needs to vary depending on the subset size, and providing the novel and efficient window selection strategy from samples ordered by the difficulty score.
>
> Regarding the CCS, CCS not only takes more time to choose the window but also requires tuning of a hyperparamter, which determines the one-sided threshold of the window, for pruning $\beta$\% hard samples. The grid search for $\beta$ not only introduces extra costs in selecting effective coresets but also requires validation or test dataset to evaluate each window subset. The authors in CCS left the efficient search of finetuning $\beta$ for future work. On the other hand, our work does not have any hyperparameter to tune for the window search, since we propose a proxy task to choose the window subset without any validation or test dataset. We think that this main difference makes our work effectively applicable in practical data subset selection scenarios.
>
> Furthermore, we’d like to highlight that the performance of BWS is higher even compared to the numbers reported in the original CCS paper obtained after finetuning $\beta$ for each selection ratio. This may come from the fact that our method prunes both easy and hard samples in the process of choosing the window subset, but CCS only prunes the hard samples and selects among the remaining samples with a uniform difficulty score distribution.
>
> | Subset ratio                              | 10%   | 20%   | 30%   | 50%   |
> |:-----------------------------------------:|:-----:|:-----:|:-----:|:-----:|
> | CCS in original paper                     | 85.7  | 90.93 | 92.97 | 95.04 |
> | Stratified only(beta 0) in original paper | 59.23 | 81.82 | 90.78 | 95.13 |
> | CCS in Ours                               | 81.56 | 89.28 | 92.5  | 94.78 |
> | BWS(Ours)                               | 88.05 | 91.29 | 93.17| 94.93 |

---

### Official Review · Reviewer_j18Y · 2023-10-31

**Soundness:** 3 good
**Presentation:** 3 good
**Contribution:** 2 fair
**Rating:** 5
**Confidence:** 3

**Summary:**

The paper presents an approach, known as Best Window Selection (BWS), designed to tackle the challenges associated with data subset selection in machine learning. BWS allows for the adaptable selection of subsets based on sample difficulty scores and consistently delivers competitive performance over a broad range of selection ratios, spanning from 1% to 90%. It excels in comparison to existing score-based and optimization-based methods when applied to datasets like CIFAR-10/100 and ImageNet.

**Strengths:**

1) The problem studied is meaningful and significant: finding a versatile data selection approach capable of sustaining competitive performance across a diverse range of selection ratios.
2) Experiments show that the proposed BWS consistently outperforms other baselines, including both score-based and optimization-based approaches.
3) The authors provide code, which enhances the reproducibility.

**Weaknesses:**

1) The notion of a "window" refers to a fixed-length interval within a sorted dataset. The "Best Window Selection (BWS)" algorithm operates under the assumption that the most optimal subset should be contiguous regarding the level of difficulty. However, the paper lacks an in-depth analysis of this particular aspect.

2) It would be intriguing to explore the broader scenario where a "window" comprises several smaller intervals and varying starting points.

3) Figure 3's readability could be enhanced by employing more distinguishable colors and markers for clarity.

**Questions:**

Kindly refer to the weaknesses.

---

> ### Author Response · Authors · 2023-11-16
> **Response to Reviewer j18Y (1/2)**
>
> We sincerely appreciate the reviewer for the feedback. Hope this response answers the reviewer’s questions.
>
> >**1. The notion of a "window" refers to a fixed-length interval within a sorted dataset. The "Best Window Selection (BWS)" algorithm operates under the assumption that the most optimal subset should be contiguous regarding the level of difficulty. However, the paper lacks an in-depth analysis of this particular aspect.**
>
> We’d like to highlight that the rationale behind the “window” selection from a sorted dataset mainly lies on two critical merits of this approach: 1) computational efficiency in searching for the optimal subset and 2) flexibility in choosing the data subset, enabling the selection of easy, moderate, or hard data subsets. In particular, this flexibility is pivotal in achieving competitive performances across a broad range of selection ratios.
>
> As pointed out by the reviewer, the optimal subset might not be necessarily strictly contiguous in terms of the difficulty level of the samples. However, our theoretical analysis in Sec. 3.1 and the ablation study in Table 3 deliver some insights on the benefit of choosing samples from the continuous interval. First of all, in the simple toy example of binary classification problem introduced in Sec. 3.1, we show that the difficulty level of the optimal subset indeed changes depending on the subset size. In particular, for sample-deficient regimes, it is better to select easy samples (that are farther from the decision boundary), while selecting difficult samples (closer to the decision boundary) is more beneficial for sample-sufficient regimes. This theoretical analysis aligns well with the empirical findings (summarized in Table 1) that the previous score-based selection methods, which select rare or difficult-to-classify samples, achieve a better performance in high selection ratios, while the coreset selection methods, which select representative samples approximating the average behavior of the total dataset, achieves a stronger performance in low selection ratios. So, we can see that the optimal subset should be composed of relatively easy (hard) samples in low (high) selection ratios.
>
> Then, the remaining question is whether there exists such a desirable difficulty level for the subset in the intermediate selection ratios. We do not provide theoretical analysis on this aspect, but present some empirical evidence in our ablation study. In particular, our ablation study in Table 3 compares different types of subsets (in addition to the hard-only or easy-only windows), such as “Hard-easy”, which combines the easy and hard samples with equal proportions or “25-75\%”, which includes random samples from a moderate regime after pruning a fixed portion of too easy or hard samples. Compared to these non-contiguous subset selection methods, our window selection consistently achieves better performance over all the selection ratios (10 to 40\%) by cleverly choosing the window starting point. This ablation study, combined with our pruning experiment results summarized in Figure 4, partly demonstrate that our window selection, which prunes top $s$\% of the most difficult samples and top $(100-w-s)$\% of the easiest samples with a properly chosen $s$ for a fixed width of $w$\%, provides a good enough flexibility in choosing a subset of competitive performances across a broad range of selection ratios.
>
> Lastly, we provide further evidence that even though our method selects samples from a contiguous range of difficulty level, our subset maintains diversity in sample selection (in the embedding space), which is often considered as a desirable criteria in subset selection. To show this, we compute the coverage (\%) of the full dataset from the $k$-nearest neighbors of the selected samples from our method (BWS), and compare this coverage with that of random selection. For example, for the subset ratio of 10% and $k=3$, if the coverage is 30\%, it means that all the selected samples do not share their neighborhoods within distance 3 in the embedding space. The tables below show that the coverage of our method is as high as that of the random selection, showing that selecting samples of similar difficulty does not reduce the sample diversity in the embedding space compared to the random selection.
>
> The portion included in the 3-nearest neighborhood (k=3)
> | Subset ratio  | 1%   | 5%    | 10%   | 20%   | 30%   |
> |:-------------:|:----:|:-----:|:-----:|:-----:|:-----:|
> | Window subset | 2.9% | 13.2% | 25.2% | 42.7% | 55.0% |
> | Random subset | 2.9% | 13.9% | 26.0% | 45.3% | 59.6% |
>
> The portion included in the 5-nearest neighborhood (k=5)
> | Subset ratio  | 1%   | 5%    | 10%   | 20%   | 30%   |
> |:-------------:|:----:|:-----:|:-----:|:-----:|:-----:|
> | Window subset | 4.8% | 19.3% | 36.5% | 56.2% | 67.4% |
> | Random subset | 4.8% | 21.7% | 38.3% | 60.7% | 74.2% |

---

> ### Author Response · Authors · 2023-11-16
> **Response to Reviewer j18Y (2/2)**
>
> >**2. It would be intriguing to explore the broader scenario where a "window" comprises several smaller intervals and varying starting points.**
>
> We appreciate the reviewer for the interesting suggestion. We conducted an additional experiment on the CIFAR-10 dataset for finding the optimal two half-width windows while varying their starting points. In detail, for a subset of size $w$\%, we searched over all combinations of two half-width windows, denoted by $[x_1, x_1+w/2] \cup [x_2, x_2+w/2]$ while varying their starting points $(x_1, x_2)$ in $x_1\in[0, 100-w]$ and $x_2\in [x_1+w/2, 100-w/2]$ with a step size of 5\%. We trained ResNet18 on each subset and checked the corresponding test accuracies. Below we report the top five results (the compositions of half-width windows and their test accuracies) for subset ratios ranging from 10 to 40\%. We highlight the cases where the two half-width windows are contiguous to each other with bold letters.
>
> We can observe that for every considered subset ratio, the top-five best performing cases include contiguous windows (or windows near to each other with the gap of only 5\%), even though we allowed flexibility in choosing the two half-width windows far away from each other. This result further supports our claim (in the response to the reviewer’s first question) that our window selection, which chooses a continuous interval of samples based on difficulty scores, successfully finds the near-optimal subset in an efficient manner across a broad range of selection ratios. The related results are added in Appendix G.3 of our revised paper.
>
> | Subset ratio |  Rank             | 1st            | 2nd            | 3rd            | 4th            | 5th            |
> |:--------------:|:---------------:|:----------------:|:----------------:|:----------------:|:----------------:|:----------------:|
> | 10%          | Half-width windows     | **35-40%, 40-45%** | **40-45%, 45-50%** | 30-35%, 45-50% | 40-45%, 50-55% | 40-45%, 55-60% |
> |          | Test accuracy | **88.05**          | **87.82**         | 87.8           | 87.74          | 87.71          |
> | 20%          | Half-width windows     | **20-30%, 30-40%** | 15-25%, 35-45% | 20-30%, 35-45% | **15-25%, 25-35%** | **25-35%, 35-45%** |
> |          | Test accuracy | **91.69**         | 91.61          | 91.56          | **91.34**          | **91.29**          |
> | 30%          | Half-width windows    | 10-25%, 30-45% | **10-25%, 25-40%** | 5-20%, 45-60%  | **15-30%, 30-45%** | 15-30%, 35-50% |
> |          | Test accuracy | 93.47          | **93.35**          | 93.31          | **93.17**          | 93.11          |
> | 40%          | Half-width windows     | 5-25%, 30-50%  | **5-25%, 25-45%**  | 0-20%, 25-45%  | 5-25%, 35-55%  | 5-25%, 40-60%  |
> |           | Test accuracy | 94.68          | **94.38**          | 94.35          | 94.23          | 94.11          |
>
>
>
>
> >**3. Figure 3's readability could be enhanced by employing more distinguishable colors and markers for clarity.**
>
> We modified the figures (Figure 3, 4 and 6) by adding markers for better readability. Thank you for your feedback.

---

### Author Response · Authors · 2023-11-16
**The revised paper is uploaded**

We sincerely thank the reviewer for their constructive feedback. Based on the reviews, we have revised and uploaded our paper with the following modifications:

* (Figure 3,4,6) We updated the plots by adding different markers for better readability.
* (Appendix B) We added a section providing some discussions on the use of the kernel ridge regression as a proxy task for evaluating the window subsets, and compared it with another proxy using the training of a simple network.
* (Appendix G.3) We provided an additional ablation experiment of different window types, composed of two half-width windows of different starting points.
* (Appendix H) We updated Table 18, which reports the fine-tuning experiemtal results on ViT.

---

### Author Response · Authors · 2023-11-21
**A gentle reminder to Reviewers**

We sincerely appreciate your time and effort in reviewing our work.
We tried our best to address the reviewers' concerns and questions in our responses.
We would like to kindly ask the reviewers to check our responses.
If you have any additional comments or questions, we would be happy to provide further responses.


Sincerely,

Authors.

---

> ### Author Response · Authors · 2023-11-22
> **A gentle reminder to Reviewers**
>
> We would like to thank the reviewers again for their constructive comments. We have uploaded the revised version and responded to all the reviewers in detail. We would greatly appreciate it if the reviewers could kindly advise if our responses solve their concerns. Thank you for your time.
>
> Sincerely,
>
> Authors.

---

### Meta-Review · Area_Chair_bFeR · 2023-12-15

**Metareview:**

This paper presents an approach, which they call, Best Window Selection (BWS), which can select data subsets of different ratios to the full dataset and can perform well for all data subset sizes. I think the main contribution of this work is being able to work well for different subset sizes compared to existing approaches.

This is a borderline paper and none of the reviewers are championing this work. I would recommend the authors take a careful look at the comments by the reviewers and submit this work to another venue. Additionally, I would also encourage the authors to compare to repeated random sampling (https://arxiv.org/abs/2305.18424) which is a very simple baseline for data subset selection yet performs surprisingly well!

**Justification For Why Not Higher Score:**

This is a borderline paper. While this paper does make several nice contributions, I feel it will be beneficial for the authors to take the suggestions by the reviewers into account and resubmit this work.

**Justification For Why Not Lower Score:**

N/A

---

### Decision · Program_Chairs · 2024-01-16

Reject